# Beyond Factuality: A Comprehensive Evaluation of Large Language Models as Knowledge Generators

**Liang Chen**[1][†] **Yang Deng**[3]**, Yatao Bian**[2][‡] **Zeyu Qin**[4]**,**
**Bingzhe Wu**[2]**, Tat-Seng Chua**[3]**, Kam-Fai Wong**[1][‡]

[1]The Chinese University of Hong Kong, [2]Tencent AI Lab
[3]National University of Singapore, [4]The Hong Kong University of Science and Technology
lchen@se.cuhk.edu.hk

## Abstract

Large language models (LLMs) outperform information retrieval techniques for downstream knowledge-intensive tasks when being prompted to generate world knowledge. Yet, community concerns abound regarding the factuality and potential implications of using this uncensored knowledge. In light of this, we introduce CONNER, a COmpreheNsive kNowledge Evaluation fRamework, designed to systematically and automatically evaluate generated knowledge from six important perspectives – *Factuality, Relevance*, *Coherence*, *Informativeness*, *Helpfulness* and *Validity*. We conduct an extensive empirical analysis of the generated knowledge from three different types of LLMs on two widely-studied knowledge-intensive tasks, *i.e.*, open-domain question answering and knowledge-grounded dialogue. Surprisingly, our study reveals that the factuality of generated knowledge, even if lower, does not significantly hinder downstream tasks. Instead, the relevance and coherence of the outputs are more important than small factual mistakes. Further, we show how to use CONNER to improve knowledge-intensive tasks by designing two strategies: Prompt Engineering and Knowledge Selection. Our evaluation code and LLM-generated knowledge with human annotations will be released[1] to facilitate future research.

## 1 Introduction

The exceptional success of large language models (LLMs) like ChatGPT and GPT4 (Ouyang et al., 2022; OpenAI, 2023) has fueled a growing interest in substituting traditional models with LLMs to attain superior performance across various NLP tasks (Liu et al., 2023b; Jagerman et al., 2023; Wang et al., 2023). In open-domain question answering (QA) and knowledge-grounded dialogue,

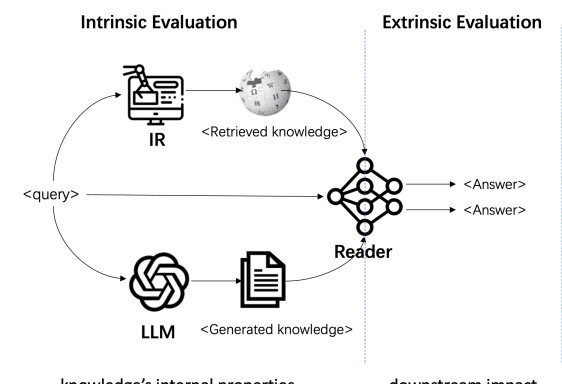

Figure 1: The CONNER Framework: Intrinsic evaluations probe the internal properties of acquired knowledge, while extrinsic evaluations assess its downstream impacts. This framework applies universally to two-stage processes in knowledge-intensive tasks.

LLMs have demonstrated superior performance than information retrieval (IR) models (Karpukhin et al., 2020) when it comes to generating world knowledge (Yu et al., 2023; Liu et al., 2022) for the downstream tasks. However, the knowledge generated may contain inherent issues, such as false statements or off-topic information. Therefore, the lack of extensive evaluation of this knowledge raises concerns about its use in downstream tasks.

To this end, four lines of research emerge. Firstly, human evaluations are conducted to assess the generated knowledge from diverse perspectives (Li et al., 2022; Yu et al., 2023; Liu et al., 2023a). However, their time-consuming nature and subjectivity often encounter issues of scalability and reproducibility. Secondly, datasets have been constructed to evaluate open-domain generation with the aid of references (Honovich et al., 2021; Glover et al., 2022a; Lee et al., 2023; Li et al., 2023). These methods, while more objective, are limited by their dependence on human-labelled references, impacting their real-world applicability and generalizability to dynamically generated content. Thirdly, self-evaluation methods (Kadavath et al., 2022b; Manakul et al., 2023) esti-

---

[†]Partial work was done in his Tencent AI Lab internship.
[‡] Corresponding author.

[1]https://github.com/ChanLiang/CONNER

| Evaluation Taxonomy | | Definition |
|---|---|---|
| *Intrinsic* | `Factuality` | whether the information in the knowledge can be verified by external evidence. |
| | `Relevance` | whether the knowledge is relevant to the user query. |
| | `Coherence` | whether the knowledge is coherent at the sentence and paragraph levels. |
| | `Informativeness` | whether the knowledge is new or unexpected against the model's existing knowledge. |
| *Extrinsic* | `Helpfulness` | whether the knowledge can improve the downstream tasks. |
| | `Validity` | whether the results of downstream tasks using the knowledge are factually accurate. |

Table 1: Taxonomy of evaluation metrics of acquired knowledge.

mate a model's uncertainty in its generated content. Despite simplicity, they lack interpretability and are less effective for long-form answers. Lastly, contemporary studies (Pan et al., 2023; Min et al., 2023) apply fact-checking principles to spot factual inaccuracies. However, these evaluation methods mainly assess a single aspect of the intrinsic quality of generated knowledge, overlooking other facets and their extrinsic impact on downstream tasks, thereby limiting a comprehensive understanding of LLM-generated content.

In light of these limitations, we propose CONNER, a COmpreheNsive kNowledge Evaluation fRamework, as illustrated in Figure 1. CONNER is designed to be a reference-free framework that can systematically and automatically evaluate the generated knowledge from six fine-grained perspectives, including diverse intrinsic evaluation of its internal properties, as well as uniform extrinsic evaluation of its impact on specific downstream tasks. The taxonomy of evaluation metrics is presented in Table 1. Based on CONNER, we conduct empirical evaluations on three different types of LLMs, including LLaMA (Wei et al., 2022) (a base LLM), FLAN-T5 (Wei et al., 2022) (an instruction-tuned LLM), Chat-GPT (Ouyang et al., 2022) (a commercial LLM trained with human feedbacks). We evaluate them on two widely-studied knowledge-intensive tasks: open-domain QA (Kwiatkowski et al., 2019) and knowledge-grounded dialogue (Dinan et al., 2018).

Our detailed investigations yield several valuable insights about the LLM-generated knowledge: 1) LLM-generated knowledge surpasses retrieved knowledge in most evaluation perspectives, while it actually suffers from the factuality issue as expected. Notably, the factuality of downstream tasks is found to be less affected by this issue, when compared to the impact of lower relevancy and coherency observed in the retrieved knowledge (§ 4.3). 2) Several critical factors are identified to influence the factuality of the generated knowledge, such as their frequency and length, while few-shot in-context learning and larger size of models do not

necessarily guarantee higher quality and reliability (§ 4.4). 3) In addition to assessing and analyzing the generated knowledge from different LLMs, the evaluation outcome of CONNER can be exploited to enhance knowledge generation and further improve the performance of downstream tasks (§ 5).

Our main contributions are as follows:
- We conduct the first empirical analysis focusing on both intrinsic quality and extrinsic reliability of the generated knowledge from LLMs.
- We propose CONNER, a COmpreheNsive kNowledge Evaluation fRamework that enables the automatic evaluation of LLMs as knowledge generators from diverse perspectives, eliminating the need for human-labelled references.
- The extensive evaluation and analysis yield profound insights and valuable practical experience for leveraging LLMs as knowledge generators.
- We collect a new set of multi-perspective human judgments of LLM-generated knowledge for two knowledge-intensive generation datasets. We demonstrate that CONNER aligns well with human judgments. The human annotations will be released to facilitate future research.

## 2 Related Work

Knowledge-intensive tasks rely heavily on access to external knowledge sources, such as open-domain dialogue and QA (Dinan et al., 2018; Kwiatkowski et al., 2019; Petroni et al., 2021). The main-streamed methods (Karpukhin et al., 2020; Lewis et al., 2020; Izacard and Grave, 2021) typically employ IR techniques to first retrieve the relevant knowledge from Wikipedia and then produce the answer or response conditioned on the knowledge. Nowadays, with the powerful capabilities of LLMs (OpenAI, 2023; Kadavath et al., 2022a), a new trending approach is to leverage LLMs to directly generate the relevant knowledge for a given query and then apply the model-generated knowledge to complete the downstream tasks (Liu et al., 2022; Li et al., 2022; Yu et al., 2023). Despite outperforming retrieval-based methods, these knowl-

edge generation techniques lack rigorous evaluation of their quality and reliability, which may contain misleading or even plausible false information, *e.g.*, hallucination and factual inconsistency.

These issues are prevalent across various NLP tasks (Ji et al., 2023). However, most studies target specific downstream tasks, such as text summarization (Maynez et al., 2020; Wang et al., 2020; Kryscinski et al., 2020a; Pagnoni et al., 2021), dialogue generation (Dziri et al., 2022; Chen et al., 2023; Xue et al., 2023; Deng et al., 2023), and fact verification (Thorne et al., 2018; Wadden et al., 2020; Schuster et al., 2021; Pan et al., 2023). These tasks are designed to examine consistency either between the input and output or between the input and a human-labeled reference, *e.g.,* the source document and its summary, the grounded knowledge and the generated response, or a human-written claim and pre-annotated references.

The success of LLMs and generative search engines (Zhao et al., 2023; Zhu et al., 2023) have brought hallucinations in LLM outputs (Rawte et al., 2023; Zhang et al., 2023) into focus. Research typically falls into four categories. (Lee et al., 2023; Li et al., 2023) aim to assess the factuality of open-domain generation automatically using specially designed datasets, but their reliance on references may limit real-world applicability. Another stream of work (Li et al., 2022; Yu et al., 2023; Liu et al., 2023a) uses human evaluation to measure output quality, which is difficult to scale. A third approach (Kadavath et al., 2022b; Manakul et al., 2023) detects hallucinations by examining the model's uncertainty or confidence, which can be inaccurate for long answers. Lastly, recent studies (Peng et al., 2023; Min et al., 2023) apply fact-checking principles to spot factual inaccuracies.

Different from previous studies, we propose a **comprehensive framework** for evaluating knowledge generated by LLMs. Our goal is to **automatically test** the intrinsic quality and extrinsic impact of generated information in knowledge-intensive tasks, *without requiring knowledge labelling or human involvement*. Through extensive testing with this framework, we aim to deepen and broaden our understanding of LLM-generated knowledge and provide valuable insights for future research.

## 3   The Evaluation Framework

We introduce CONNER, a comprehensive and innovative framework, specifically designed for the rigorous evaluation of the quality and dependability of knowledge used in knowledge-intensive tasks. CONNER is rooted in in-depth error analysis, paving the way for the construction of an evaluation taxonomy, which integrates six unique perspectives into two coherent categories, as delineated in Table 1. Capitalizing on the advantages of unsupervised metrics, our framework eliminates the need for human-labeled reference knowledge and standardizes scores within an intuitive range of $[0, 1]$, simplifying comparison and interpretation.

The subsequent subsections provide a detailed examination of the framework's design, commencing with the formulation of knowledge-intensive tasks and the identification of associated error patterns. These insights direct the design of our metrics. Through comprehensive intrinsic and extrinsic evaluations, we aim to gain a holistic understanding of the LLMs-generated knowledge.

### 3.1   Tasks Formulation

Formally, we define the knowledge-intensive task as follows: given a user query $q$, the goal is to produce an answer with access to knowledge resources as illustrated in Figure. 1. Specifically, the system first obtains the relevant knowledge $k$ that can help answer the query $q$ from knowledge resources $\mathcal{K}$, then the reader generates an answer $a$ using the acquired knowledge $k$. Specifically, the knowledge resource $\mathcal{K}$ can be either a knowledge base for knowledge retrieval or language models for knowledge generation. Detailed formulations of these two settings are presented in Appendix A.

### 3.2   From Error Patterns to Metrics Design

To identify common errors by LLMs in knowledge-intensive tasks and create a more targeted evaluation framework, we used thematic analysis (Braun and Clarke, 2012). We began by extracting and consolidating patterns from subtle errors in knowledge and answers in responses from LLaMA to 160 samples from NQ (Kwiatkowski et al., 2019) and WoW (Dinan et al., 2018) datasets. To ensure the breadth of the error spectrum was adequately represented, we further substantiated these patterns using additional questions from NQ and WoW. As a result, we discerned four primary error categories in knowledge generation and two in answer generation. In response, we devised four intrinsic metrics for knowledge evaluation and two extrinsic metrics for answer evaluation, as outlined in Table 1.

## 3.3 Intrinsic Evaluation

Intrinsic evaluation refers to the assessment of the acquired knowledge based on its internal properties and performance, without considering its impact on downstream tasks or applications. In specific, we implement four model-based metrics for evaluating the acquired knowledge in terms of *factuality*, *relevance*, *informativeness*, and *coherence*.

**Factuality** The core of factuality assessment is validating the acquired knowledge by external evidence [2]. Given an acquired knowledge $k = \{s_1, \ldots, s_m\}$ composed of $m$ sentences, we can use a dense retrieval model (Santhanam et al., 2021) or search engine API to recall the $l_i$ most relevant evidence $E_i = \{e_{i,1}, \ldots, e_{i,l_i}\}$ for each sentence $s_i$ from the expert knowledge base or the internet. After collecting all the evidence $E = \{E_1, \ldots, E_m\}$, the factuality score is computed as follows:

$$\mathbf{S}_{\mathsf{fact}}(k, E) = \min_{i=1..m} f(s_i, E_i)$$
$$= \min_{i=1..m} \max_{j=1..l_i} \mathrm{NLI}(s_i, e_{i,j}) \quad (1)$$

where $f(\cdot)$ is a function to compute sentence-level factuality, $\mathrm{NLI}(\cdot)$ is a natural language inference model processing a premise-hypothesis pair to output a $R^3$ vector, indicating whether a hypothesis ($s_i$) is entailed by, neutral to or refuted by the given premise ($e_{i,j}$). Following these computations, sentence-level results are aggregated along the entailment dimension using one of three operations: $\min$, $\mathrm{mean}$, or $\max$ to match the desired error tolerance level. In this instance, we exemplify the process using $\min$. Finally, we obtain a three-dimensional factuality score $\mathbf{S}_{\mathsf{fact}}(k, E)$. From each dimension of this vector, we can derive three fine-grained scores. We denote those scores as `factual-consistent`, `non-verified`, and `factual-inconsistent`, respectively.

This strategy seeks to address the shortcomings of traditional factuality metrics (Wang et al., 2020; Honovich et al., 2021; Glover et al., 2022a; Lee et al., 2023) that mainly depend on consistency with human-annotated references. These metrics often fail in emerging knowledge generation scenarios (Table 10), as they struggle with model-generated content beyond reference knowledge scope and face difficulties when references are unavailable in real-world applications. Our method of evidence collection and results aggregation effectively tackles these issues.

---

[2] We empirically demonstrate ground-truth knowledge is dispensable for the factuality evaluation in Appendix B.

**Relevance** To assess the relevance between a given query $q$ and the acquired knowledge $k$, we compute the relevance score as follows:

$$S_{\mathsf{rel}}(k, q) = \mathrm{Matching}(k, q) \quad (2)$$

The $\mathrm{Matching}(\cdot)$ function denotes a fine-grained matching model specifically designed for assessing the relevance between the query and knowledge. In our study, we employ the BERT ranking model (Nogueira et al., 2019) for this purpose.

This methodology addresses the limitations that arise when traditional relevance metrics are applied within knowledge generation scenarios. Traditional relevance metrics (Karpukhin et al., 2020; Shuster et al., 2021; Komeili et al., 2021), which typically rely on word overlap or similarity with human-written references, face two significant challenges. First, these traditional metrics do not correspond well with scenarios where LLMs serve as generative search engines, as evidenced by the unsatisfactory results in Table 10. Second, the reliance on reference knowledge constitutes a substantial challenge, especially when such references are scarce or absent in real-world applications. Contrarily, our BERT ranking model, trained on manually annotated Bing search data, excels at comparing the relevance of different knowledge to a given query.

**Coherence** As the acquired knowledge is typically long-form texts composed of multiple sentences, we propose to measure sentence-level cohesion and paragraph-level coherence: the former measures the cohesion of individual sentences, and the latter measures the coherence between sentences. The sentence-level cohesion score $S_{\mathsf{coh\_sent}}(k)$ is computed as follows:

$$S_{\mathsf{coh\_sent}}(k) = \frac{1}{m} \sum_{i=1}^{m} 1/\mathrm{PPL}(s_i) \quad (3)$$

where $\mathrm{PPL}(\cdot)$ is computed by a GPT-based model (Radford et al., 2019; Black et al., 2021), measuring the perplexity for each sentence.

On the other hand, the paragraph-level coherence score is determined by the normalized score of a discourse coherence model (Jwalapuram et al., 2021), denoted as $S_{\mathsf{coh\_para}}(k)$:

$$S_{\mathsf{coh\_para}}(k) = \mathrm{Scorer}_{\mathrm{para}}(s_1, ..., s_m) \quad (4)$$

By considering both sentence-level cohesion and paragraph-level coherence, we gain insights into the overall coherence of the acquired knowledge.

**Informativeness** To assess the informativeness of the procured knowledge—defined as the degree to which the knowledge is novel or unexpected in relation to the model's existing knowledge about the query—we calculate the informativeness score of the acquired knowledge $k$ given $q$ as follows:

$$S_{\text{info}}(k, q) = 1 - \exp\left(\frac{1}{M}\sum_{t=1}^{M}\ln P_\theta(k_t|k_{1:t-1}, q)\right) \quad (5)$$

Assuming the unbiased benchmark model $\theta$ encapsulates world knowledge from general pretraining data, we thus select the GPT-2 series models.

To grasp the expected behaviour of this metric, consider a simple query: *"What is the capital of the United States?"* The knowledge acquired here is *"Washington"*. In this situation, the model's average probability of generating *"Washington"* is high, as it already knows this fact. Consequently, our informativeness score for this knowledge would be low. Conversely, if the acquired knowledge was *"Chicago"*, the model's probability of generating it would be low. This knowledge is surprising compared to its existing knowledge, resulting in a high informativeness score. On the other hand, for a tough query where the model is clueless, any provided knowledge would score high on informativeness due to the model's low output probabilities.

### 3.4 Extrinsic Evaluation

Extrinsic evaluation, in contrast to intrinsic evaluation, focuses on uniformly assessing the performance of the acquired knowledge within the context of different downstream tasks. Specifically, we measure how well the acquired knowledge contributes to the downstream task on two types of metrics (*helpfulness* and *validity*). Extrinsic evaluation provides a more comprehensive understanding of the practical value of the acquired knowledge.

**Helpfulness** Given a query and answer pair $(q, a)$, we assess to what extent the acquired knowledge $k$ can help answer the query. As we assume no pre-annotated ground-truth knowledge, we use irrelevant knowledge as the baseline. Specifically, we randomly sampled $u$ knowledge $\{k_1^-, \cdots, k_u^-\}$ to reduce the variance of baseline estimation. Then the helpfulness score is computed as follows:

$$S_{\text{help}}(q, a, k, k_1^-, \cdots, k_u^-)$$
$$= \max(0, 1 - \frac{\mathcal{L}(q, k, a)}{\frac{1}{u}\sum_{i=1}^{u}\mathcal{L}(q, k_i^-, a)}) \quad (6)$$
$$= \max(0, 1 - \frac{\log P(a|q, k)}{\frac{1}{u}\sum_{i=1}^{u}\log P(a|q, k_i^-)})$$

where $\mathcal{L}(q, k, a)$ and $\mathcal{L}(q, k_i^-, a)$ are cross entropy losses of answer generation using $k$ and $k_i^-$ respectively. Ideally, the generated knowledge $k$ can provide enough information and reduce the $\mathcal{L}(q, k, a)$ to zero, and then the helpfulness score equals one. The worst case is the generated knowledge is no better than random knowledge $(\mathcal{L}(q, k, a) \geq \frac{1}{u}\sum_{i=1}^{u}\mathcal{L}(q, k_i^-, a))$, and the helpfulness score is naturally zero.

**Validity** To measure how the reliability of the acquired knowledge affects the factuality of the generated answer $a$ on downstream tasks, we define the validity metric for two types of downstream tasks: span-based answers (*e.g.*, open-domain QA) and open-ended answers (*e.g.*, knowledge-grounded dialogue). As for span-based answers, the generated answers cannot form a complete sentence for factuality measurement. To this end, we concatenate $(q, a^*)$ as the premise and $(q, a)$ as the hypothesis for deriving the `factual-consistent` score of the $\text{NLI}(\cdot)$ model as the validity score:

$$S_{\text{val}}(q, a^*, a) = \text{NLI}_{\text{fact}}((q, a), (q, a^*)) \quad (7)$$

where $a^*$ denotes the ground-truth answer for downstream tasks and the $\text{NLI}(\cdot)$ model is the same as that of Eq. (1).

We demonstrate this measure outperforms traditional metrics like Exact Match and F1 score as shown in Table 10, which rely on literal matches, and often yield low recall. For instance, an entity pair like 'PRC' and 'China' would receive a zero score due to their differing literal presentations.

As for open-ended answers, we collect $l$ evidence $E = \{e_1, \ldots, e_l\}$ and adjust Eq. (1) to be:

$$S_{\text{val}}(a, E) = f(a, E) = \max_{i=1..l}\text{NLI}_{\text{fact}}(a, e_i) \quad (8)$$

## 4 Evaluation

In this section, we will first validate our proposed metrics, and then leverage them to comprehensively evaluate three different types of LLMs across two knowledge-intensive tasks, followed by an in-depth analysis of the results.

### 4.1 Metrics Efficacy Validation

To validate the effectiveness of our proposed metrics, we conducted manual evaluations and compared the results with baseline metrics. Specifically, we developed specific annotation guidelines for each metric, detailed in Appendix J, and performed manual annotations accordingly. These

| Model | Setting | Factuality | | | Relevance | Coherence | | Inform. | Helpful. | Validity |
|---|---|---|---|---|---|---|---|---|---|---|
| | | Fact-cons. | Non-verif. | Fact-incon. | | Coh-sent. | Coh-para. | | | |
| DPR | Supervised | **97.78%** | 2.23% | 0.00% | 0.7514 | 0.0301 | 0.7194 | **0.8965** | 0.1236 | 36.86% |
| FLAN-T5 | Zero-shot | 58.40% | 27.80% | 13.80% | 0.6848 | **0.1249** | 0.7776 | 0.6727 | 0.0000 | 32.47% |
| LLAMA | | 94.20% | 4.80% | 1.00% | 0.7316 | 0.1183 | 0.8240 | 0.7572 | 0.2191 | 42.00% |
| CHATGPT | | 83.63% | 13.6% | 2.77% | 0.8491 | 0.0909 | **0.9033** | 0.7330 | 0.1461 | **43.35%** |
| FLAN-T5 | Few-shot | 20.75% | 62.40% | 25.40% | 0.6787 | 0.0416 | 0.8110 | 0.6899 | 0.0000 | 34.65% |
| LLAMA | | 89.00% | 9.20% | 1.80% | 0.6966 | 0.0776 | 0.8550 | 0.8545 | **0.2528** | 40.49% |
| CHATGPT | | 86.07% | 10.97% | 2.96% | **0.9205** | 0.0653 | 0.8837 | 0.7700 | 0.1966 | 42.36% |

Table 2: Automatic evaluation results of different LLMs in the Natural Question test set. Underlined and **Bold** results denote the best results among each setting and among all settings, respectively.

| Model | Setting | Factuality | | | Relevance | Coherence | | Inform. | Helpful. | Validity |
|---|---|---|---|---|---|---|---|---|---|---|
| | | Fact-cons. | Non-verif. | Fact-incon. | | Coh-sent. | Coh-para. | | | |
| DPR | Supervised | **91.96%** | 5.18% | 2.87% | 0.0907 | 0.0223 | 0.6569 | **0.9357** | 0.0000 | 61.52% |
| FLAN-T5 | Zero-shot | 77.90% | 17.28% | 4.82% | 0.3776 | 0.1203 | 0.8331 | 0.7239 | 0.0904 | 56.97% |
| LLAMA | | 89.46% | 8.89% | 1.65% | 0.5041 | 0.0548 | 0.8389 | 0.7889 | **0.1178** | 63.50% |
| CHATGPT | | 88.51% | 10.38% | 1.11% | **0.5283** | 0.1028 | **0.9250** | 0.7448 | 0.1023 | 59.76% |
| FLAN-T5 | Few-shot | 76.50% | 17.20% | 6.30% | 0.4463 | **0.1523** | 0.7988 | 0.6983 | 0.0934 | 57.18% |
| LLAMA | | 85.07% | 12.05% | 2.88% | 0.3930 | 0.1088 | 0.7947 | 0.7855 | 0.1132 | **63.79%** |
| CHATGPT | | 85.75% | 12.01% | 2.24% | 0.4618 | 0.0979 | 0.8632 | 0.7922 | 0.1164 | 60.27% |

Table 3: Automatic evaluation results of different LLMs in the Wizard of Wikipedia test set.

annotations allowed us to calculate the correlation between each metric and human evaluations. Subsequently, we compared these correlations with baseline metrics (Table 10). Our metrics demonstrated a strong correlation with human evaluations, significantly outperforming the baseline metrics. Details are presented in Chapter 6 and Appendix J.

## 4.2 Experimental Setups

**Baselines** Compared with a popular retrieval-based model, DPR (Karpukhin et al., 2020), we evaluate knowledge generation with the three different types of LLMs, including FLAN-T5 (Wei et al., 2022), LLaMA (Touvron et al., 2023), and Chat-GPT (Ouyang et al., 2022). By default, we report the results with the largest size of each LLM and adopt greedy decoding in our experiments for reproducibility. Details are presented in Appendix C.

**Datasets** We evaluate the generated knowledge on two widely-studied benchmark datasets, including 1) **Natural Questions (NQ)** (Kwiatkowski et al., 2019), an open-domain QA dataset; and 2) **Wizard of Wikipedia (WoW)** (Dinan et al., 2018) a knowledge-grounded dialogue dataset. During experiments, we randomly sample 500 examples from the NQ and WoW test sets respectively for evaluation. Details are presented in Appendix D.

**Implementation Details** All the adopted models in CONNER are introduced in Appendix E.

**Evaluation Setting** Following (Yu et al., 2023), we evaluate the knowledge generation of LLMs under both zero-shot and few-shot settings. After the knowledge acquisition, we perform QA or dialogue generation under the few-shot setting to further investigate the impact of different knowledge acquisition methods on downstream tasks.

*1) Zero-shot Evaluation*: We test with varied prompts and report peak performance. A prompt could be "Generate Wikipedia knowledge for the query. {query}". Prompts tried are in Appendix F.

*2) Few-shot Evaluation*: We construct the prompt with $k$ randomly chosen samples from the training set. The example templates used for knowledge generation are listed in Appendix F and G.

## 4.3 Overall Evaluation

Table 2 and Table 3 summarize the evaluation results of DPR and three LLM-based knowledge generators on NQ and WoW datasets, respectively. There are several notable observations as follows:

**Generated knowledge exceeds retrieved knowledge in most evaluation perspectives, except the *factuality* and *informativeness*.** In both NQ and WoW scenarios, LLMs show remarkable capabilities in generating highly *relevant* and *coherent* knowledge. Moreover, the knowledge generated by LLMs also proves to be more beneficial for downstream tasks, regarding both *helpfulness* and *validity*. These results highlight the significant ad-

| Model | Query | Acquired Knowledge | Answer |
|-------|-------|-------------------|--------|
| LLaMA | Who wrote the song the glory of love? | "The Glory of Love" is a song written by Billy Hill, recorded by American singer Benny Goodman in 1936. The song has been covered by many other artists, including a version by American singer Peggy Lee with Bing Crosby and the Johnny Mann Singers in 1959. | Billy Hill ✓ |
| FLAN-T5 | Who wrote the song the glory of love? | The Glory of Love is a song written by the American composer and lyricist Irving Berlin. | Irving Berlin ✗ |
| DPR | Rain sleet or snow that contains a high concentration of acids is called? | ... Ice pellets are a form of precipitation consisting of small, translucent balls of ice. This form of precipitation is also referred to as ""sleet"" by the United States National Weather Service. (In British English ""sleet"" refers to a mixture of rain and snow)... | icy rain ✗ |

Table 4: Factuality of acquired knowledge may not influence the validity of the answer. Red words represent factual errors in critical information, while blue words represent factual errors in non-critical information.

| Model | Extrinsic | Instrinsic | | | | |
|-------|-----------|------|------|-----------|-----------|------|
| | | Fact. | Rel. | Coh-sent. | Coh-para. | Info. |
| DPR | helpful. | 0.10 | **0.24**$^\dagger$ | 0.07 | -0.03 | -0.14$^\dagger$ |
| | validity | 0.04 | **0.19**$^\dagger$ | 0.04 | -0.06 | -0.09 |
| LLMs | helpful. | **0.14** | -0.05 | 0.10 | -0.09 | -0.05 |
| | validity | **0.15**$^\dagger$ | -0.02 | 0.07 | -0.03 | -0.03 |

Table 5: The Somers' correlation between intrinsic and extrinsic metrics on NQ. Scores with $p\text{-}value < 0.05$ are marked with $^\dagger$. **Bold** results denote the most correlated intrinsic metric to the concerned extrinsic metric. The breakdowns of all correlations are in Appendix H.

| Model | Extrinsic | Instrinsic | | | | |
|-------|-----------|------|------|-----------|-----------|------|
| | | Fact. | Rel. | Coh-sent. | Coh-para. | Info. |
| DPR | helpful. | 0.01 | **0.27**$^\dagger$ | 0.10$^\dagger$ | -0.03 | -0.14$^\dagger$ |
| | validity | -0.01 | -0.06 | **0.13**$^\dagger$ | -0.12$^\dagger$ | -0.13$^\dagger$ |
| LLMs | helpful. | 0.06 | 0.05 | **0.10** | 0.00 | -0.16 |
| | validity | **0.24**$^\dagger$ | 0.09 | 0.05 | -0.02 | -0.07 |

Table 6: The Somers' correlation between intrinsic and extrinsic metrics on WoW.

vantages of utilizing LLMs as knowledge generators in terms of knowledge quality and applicability, rendering them a valuable knowledge resource for various knowledge-intensive applications.

**Despite obtaining lower *factuality* than retrieved knowledge, generated knowledge contributes more to the factuality of downstream tasks (*i.e.*, higher *validity*).** To investigate the underlying reason, we analyze the correlation between different intrinsic metrics and extrinsic metrics on two tasks. As shown in Tables 5 and 6, the performance of downstream tasks is indeed hindered by the issue of *factuality* in the generated knowledge from LLMs. However, for retrieval models (*e.g.*, DPR), limitations may arise from the *relevance* and *coherence* of the retrieved knowledge, while its high factuality fails to ensure the performance of downstream tasks. We present a case study in Ta-

ble 4, which intuitively shows that the presence of factual errors in non-critical information has minimal impact on downstream tasks, while it is highly impossible to derive the correct answer from the irrelevant retrieved knowledge. While LLaMA and ChatGPT generate knowledge with slightly lower factuality than DPR, it is shown to be adequate for downstream tasks. At this point, the relevance of the acquired knowledge is more critical. Hence, relying solely on the factuality of the knowledge itself is an unreliable means of assessing its impact on the factuality of downstream tasks. Motivated by this finding, we investigate approaches to guiding the generated knowledge selection with the multi-perspective evaluation outcome of CONNER for improving the downstream performance in § 5.

**DPR falls short of retrieving relevant and helpful knowledge for knowledge-grounded dialogues.** As the DPR model is finetuned on QA datasets to match a question to Wikipedia knowledge, the DPR model struggles to match dialogue utterances with the necessary knowledge. Also, the candidate Wikipedia passages in DPR (100 tokens) are much longer than the knowledge needed in WoW, containing much redundant information. This reveals the shortcomings of supervised dense retrieval models, such as limited transferability and being constrained by knowledge bases.

**Few-shot in-context learning for LLMs generally harms the *factuality* of generated knowledge.** We observe that the length of knowledge generated by few-shot ICL is generally longer than that of zero-shot prompting since the ground-truth knowledge for demonstrations is relatively long. Consequently, LLM is more error-prone (see the analysis of **long-form generation** in § 4.4). This indicates that few-shot ICL is not always better than zero-shot ICL in knowledge generation, and the selected demonstrations attach great importance.

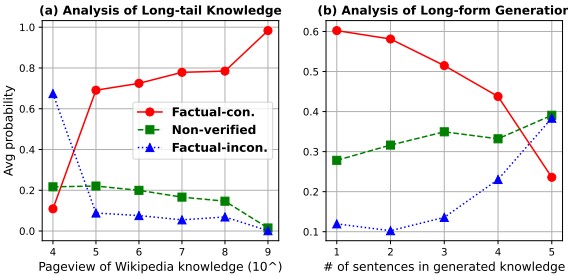

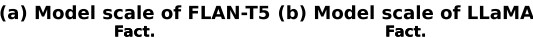
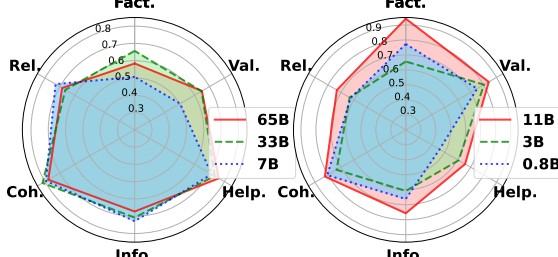

Figure 2: The impact of knowledge frequency and length on the factuality of the generated knowledge.

Inspired by this, we investigate approaches to guiding the few-shot demonstration selection with the evaluation outcome of CONNER for improving the performance of few-shot ICL in § 5.

**FLAN-T5 fails to be a qualified knowledge generator since its generated knowledge is *poorly factual* and *rarely helpful* to downstream tasks.** Although FLAN-T5 (11B) significantly surpasses many models of the same scale through instruction tuning on numerous tasks, it falls short of being a qualified knowledge generator. As shown in Table 4, such a low factuality leads to frequent occurrences of factual errors in critical information, thereby harming downstream tasks. To this end, we study the scaling of performance w.r.t different perspectives by varying the **model size** in § 4.4.

### 4.4 Further Analysis

We further analyze how different factors affect the quality and reliability of the generated knowledge and discuss our findings below.

**Long-tail Knowledge**   We investigate the impact of the knowledge frequency on the factuality performance of LLaMA on the WoW dataset. Each data entry in WoW comprises a topic, query, knowledge, and answer. The topic indicates the corresponding Wikipedia page linked to the knowledge. We assess this knowledge's frequency using Wikipedia pageviews from 2015 to 2021[3]. This enables us to differentiate between common and long-tail knowledge in WoW. Our findings reveal that LLaMA exhibits lower reliability when it is expected to generate rare/long-tail knowledge compared to common knowledge, as depicted in Figure 2(a).

**Long-form Generation**   We investigate the impact of generation length on the factuality of the generated knowledge. Specifically, we consider knowledge over 40 tokens and take sentences as evaluation units aligned with factuality evaluation.

---

[3] https://wikimedia.org/api/rest_v1

Figure 3: Performance on NQ with different sizes of FLAN-T5 and LLaMA as the knowledge generator (Help. and Val. scores are linearly scaled).

Figure 2(b) displays the factuality performance based on the number of sentences in the generated knowledge. The results show that LLaMA exhibits higher error rates when generating long-form knowledge. Therefore, prompting the LLMs to generate the required knowledge in a concise rather than lengthy manner can benefit factuality.

**Impact of Model Size**   Figures 3 depicts the performance scaling with the model size, including LLaMA-65B/33B/7B and FLAN-T5-11B/3B/780M. The results are reported on the NQ dataset using zero-shot prompting. We observe that larger models do not necessarily outperform smaller models in terms of intrinsic evaluation (particularly when parameter magnitudes are similar). However, larger models consistently outperform smaller models in terms of extrinsic evaluation(helpfulness and validity). Detailed tables are presented in Appendix I.

## 5 Two Use Cases of CONNER

To explore how our framework can guide the future design of utilizing LLMs as a knowledge generator, we design two strategies to employ CONNER as a measurement for guiding the **Prompt Engineering** and **Knowledge Selection** for knowledge-intensive tasks. We define the overall quality of knowledge $k$ given the query $q$ as follows:

$$Q_{\text{know}}(q, k) = \gamma^{\mathsf{T}} \cdot \mathbf{S}_{\text{intr}} \quad \gamma \in \mathbb{R}^4$$
$$\mathbf{S}_{\text{intr}} = [S_{\text{fact}}, S_{\text{rel}}, S_{\text{coh\_para}}, S_{\text{info}}]^{\mathsf{T}} \quad (9)$$

where $Q_{\text{know}}$ is the linear combination of four instinct metrics $\mathbf{S}_{\text{intr}}$ and $\gamma$ is the coefficient vector.

**Prompt Engineering**   We show how to use CONNER to improve knowledge generation by performing prompt engineering for few-shot ICL. We random sample a small set of $m$ samples from the training set, then use $Q_{\text{know}}(q, k)$ as the scoring function to select the top $n$ samples to compose the few-shot prompt. As shown in Table 7, the

| Model | Fact. | Rel. | Coh. | Info. |
|---|---|---|---|---|
| ChatGPT | 85.8% | 0.462 | 0.863 | **0.792** |
| ChatGPT$_{select\ prompt}$ | **87.7%** | **0.503** | **0.899** | 0.775 |

Table 7: CONNER-guided demonstration selection improves the intrinsic quality of generated knowledge.

| Model | Helpfulness | Validity |
|---|---|---|
| ChatGPT | 0.1461 | 43.45% |
| ChatGPT$_{select\ knowledge}$ | **0.2090** | **44.28%** |

Table 8: CONNER-guided knowledge selection improves extrinsic (downstream) performance.

knowledge generated by CONNER-enhanced few-shot prompting outperforms that with random demonstrations on 3 out of 4 perspectives, under the setting of $m = 30$ and $n = 8$.

**Knowledge Selection**  We employ CONNER to improve downstream tasks by selecting high-quality generated knowledge. Specifically, we generate $r$ different knowledge $\mathcal{H} = \{\tilde{k}_1, ..., \tilde{k}_r\}$ from LLMs with top-$p$ sampling, then select the generated knowledge for the downstream task, according to $k = \mathrm{argmax}_{\tilde{k} \in \mathcal{H}} Q_{\mathsf{know}}(q, \tilde{k})$. As shown in Table 8, we achieve a relative improvement of 43.15% in helpfulness on ChatGPT with $p = 0.9$ and $r = 5$.

# 6  Human Evaluation

We conducted a human evaluation by randomly selecting 400 samples from the NQ and WoW test sets. Our three annotators provided ratings for the intrinsic and extrinsic metrics for the four models. Additionally, for FLAN-T5 and LLaMA, we annotated the specific locations of factual errors in the generated knowledge, aiming to facilitate future research on fine-grained fallacy detection. Detailed annotation instructions and the statistics of our labelled data can be found in Appendix J.1.

To evaluate how well CONNER matches human evaluation of knowledge and compares with several baseline metrics, we measure the Somers' D correlation (Somers, 1962) between the human rating 0, 1, 2 of the knowledge quality and corresponding metric scores. Table 9 and Table 10 illustrate the results of four models on the NQ dataset. We observe that: (1) CONNER yields consistently good correlations with human evaluation w.r.t different evaluation perspectives (except for informativeness), which indicates that the quality of knowledge can be more effectively evaluated with CONNER. The inconsistency between informativeness and human judgment is attributed to the differences in model

| Metric | DPR | FLAN-T5 | LLaMA | ChatGPT |
|---|---|---|---|---|
| Factuality | 0.65$^{\dagger}$ | 0.66$^{\dagger}$ | 0.66$^{\dagger}$ | 0.63$^{\dagger}$ |
| Relevance | 0.69$^{\dagger}$ | 0.37$^{\dagger}$ | 0.55$^{\dagger}$ | 0.54$^{\dagger}$ |
| Coherence | 0.53$^{\dagger}$ | 0.58$^{\dagger}$ | 0.44$^{\dagger}$ | 0.49$^{\dagger}$ |
| Informative | 0.30$^{\dagger}$ | 0.17 | 0.35 | 0.32$^{\dagger}$ |
| Helpfulness | 0.75$^{\dagger}$ | 0.45$^{\dagger}$ | 0.81$^{\dagger}$ | 0.69$^{\dagger}$ |
| Validity | 0.83$^{\dagger}$ | 0.73$^{\dagger}$ | 0.85$^{\dagger}$ | 0.82$^{\dagger}$ |

Table 9: Somer's D correlation of metrics with the human annotation on NQ (The results on WoW are presented in Appendix J.2). Correlation scores with $p\text{-}value < 0.05$ are marked with $^{\dagger}$.

| Metric | DPR | FLAN-T5 | LLaMA | ChatGPT |
|---|---|---|---|---|
| Factuality | 0.65$^{\dagger}$ | 0.66$^{\dagger}$ | 0.66$^{\dagger}$ | 0.63$^{\dagger}$ |
| HE | -0.24 | 0.15 | -0.03 | 0.29$^{\dagger}$ |
| NLI | 0.23 | 0.47$^{\dagger}$ | 0.27$^{\dagger}$ | 0.38$^{\dagger}$ |
| NLI-Multitask | 0.18$^{\dagger}$ | 0.51$^{\dagger}$ | 0.26$^{\dagger}$ | 0.32$^{\dagger}$ |
| NLI-Decompose. | 0.23$^{\dagger}$ | 0.47$^{\dagger}$ | 0.27$^{\dagger}$ | 0.38$^{\dagger}$ |
| Relevance | 0.69$^{\dagger}$ | 0.37$^{\dagger}$ | 0.55$^{\dagger}$ | 0.54$^{\dagger}$ |
| F1 | 0.45$^{\dagger}$ | 0.21 | 0.41$^{\dagger}$ | 0.47$^{\dagger}$ |
| Validity | 0.83$^{\dagger}$ | 0.73$^{\dagger}$ | 0.85$^{\dagger}$ | 0.82$^{\dagger}$ |
| EM | 0.59$^{\dagger}$ | 0.51$^{\dagger}$ | 0.54$^{\dagger}$ | 0.61$^{\dagger}$ |
| F1 | 0.74$^{\dagger}$ | 0.67$^{\dagger}$ | 0.76$^{\dagger}$ | 0.77$^{\dagger}$ |

Table 10: Comparing CONNER with reference-reliant baseline metrics on the NQ dataset. Details of baseline metrics are presented in Appendix J.3.

knowledge and human knowledge. (2) CONNER metrics consistently outperform all other reference-reliant metrics, indicating the effectiveness of our framework in the knowledge evaluation scenarios.

# 7  Conclusion

In this work, we introduce CONNER, a comprehensive evaluation framework designed to automatically assess both the intrinsic quality and extrinsic reliability of the knowledge generated by LLMs. Notably, CONNER is reference-free but demonstrates a better correlation with human judgement compared with previous reference-reliant metrics.

Through extensive evaluation and in-depth analysis, we identify several key factors affecting the factuality of generated knowledge. We find although the generated knowledge is less factual than the retrieved knowledge, it remarkably enhances the factuality of downstream tasks over the retrieved ones. Furthermore, we propose two approaches to improve knowledge generation and downstream task performance with the guidance of CONNER. We believe our framework and findings will facilitate the future research of trustworthy AIGC.

## Limitations

In this section, we discuss the limitations in this work from three perspectives.

Firstly, the knowledge we evaluate primarily relies on information sourced from Wikipedia. This choice is driven by two considerations: (1) Large language models (LLMs) are trained on diverse corpora, which may include undisclosed domain-specific or task-specific data. To ensure fairness in our evaluations and enable meaningful comparisons, we focus on the common data sources that all models have learned from, with Wikipedia being a prevalent pre-training corpus for different LLMs. (2) Wikipedia is renowned for its high-quality knowledge, providing us with authoritative evidence to validate the generated knowledge. Additionally, leveraging such authoritative evidence enhances the interpretability of our factual judgments. In future work, we aim to expand our evaluations to include a broader range of world knowledge, thus further enhancing the scope and generalizability of our findings.

Secondly, while our work primarily aims to propose a general framework that can be applied to any language, our evaluation framework presents potential generalization challenges for non-English languages. This is due to its reliance on several common NLP components, a limitation echoed across many NLP methodologies. Encouragingly, the development of model variants in other languages, such as Chinese (Hu et al., 2020; Xie et al., 2023; Huang et al., 2017), indicates the potential for broader applications. Nonetheless, the reality remains that for very low-resource languages without existing NLP models, these components may need to be developed from scratch. This issue represents a challenge that the community needs to address in the future.

A third limitation is that our assessment of factuality is limited to sentence-level granularity. Through analysis and manual annotation, we have identified that large language models (LLMs) tend to exhibit errors at a more detailed level, particularly concerning numbers, time, and the generation of misleading or fabricated concepts (e.g., key characters, identities, and locations), particularly within parallel structures. To address this limitation, future research will concentrate on developing more fine-grained methods for detecting hallucinations and assessing factual accuracy. To facilitate such research, we have annotated a specific subset of data that targets fine-grained factual errors.

Despite these limitations, we believe our work serves as a significant catalyst for the automated evaluation of knowledge generated by large language models, contributing positively to the advancement of more trustworthy AI systems.

## Acknowledgements

We extend our sincerest gratitude to Professor Jing Ma, whose insightful discussions and suggestions on factuality evaluation have significantly inspired our design. We are particularly grateful to our three anonymous reviewers, whose thorough and meticulous reviews have considerably improved the quality of our work. Their constructive discussions and insights have undoubtedly enhanced our revisions. This research work is partially supported by CUHK under Project No. 3230377 (Ref. No. KPF23GW20).

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

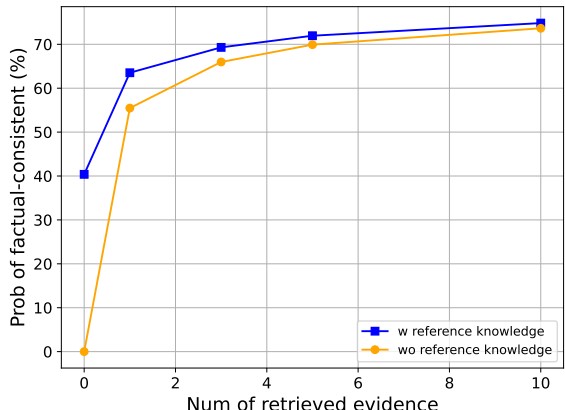

Figure 4: The influence of reference knowledge (e.g., the annotated Wiki. document in WoW dataset) in factuality evaluation weakens as the amount of retrieved evidence increases.

# Appendix

# A  Details of Problem Formulation

We provide a formulation of the two-step process for knowledge-intensive tasks, as illustrated in Fig.1. Formally, the knowledge-intensive generation problem can be formulated as the following chain rule:

$$P(a|q, \mathcal{K}) = \sum_k P(k|q, \mathcal{K})P(a|q, k) \qquad (10)$$

where $P(k|q, \mathcal{K})$ is the knowledge acquisition process and $P(a|q, k) = \prod_{t=1}^{N} P(a_t|a_{1:t-1}, q, k)$ is the autoregressive answer generation process of the reader model based on the acquired knowledge.

Retrieval-based knowledge acquisition methods use a retrieval model to retrieve the most relevant knowledge from the knowledge resource $\mathcal{K} = \{d_1, d_2, \ldots, d_K\}$ composed of $K$ documents:

$$P(k = d_i|q, \mathcal{K}) = \frac{e^{\text{sim}(q, d_i)}}{\sum_{j=1}^{K} e^{\text{sim}(q, d_j)}} \qquad (11)$$

where $\text{sim}(\cdot)$ function is used to measure the similarity, e.g., cosine similarity, between the query and the knowledge document.

Generation-based knowledge acquisition methods prompt a large language model to directly generate the required knowledge:

$$P(k|q, \mathcal{K}) = \prod_{t=1}^{M} P_{\mathcal{K}}(k_t|k_{1:t-1}, q, \texttt{prompt}) \qquad (12)$$

where prompt denotes the zero-shot or few-shot prompt and the LLM is regarded as the knowledge resource $\mathcal{K}$ and $P_{\mathcal{K}}$ stands for the distribution induced by the LLM.

| Dataset | Prompts | Best |
|---------|---------|------|
| NQ | Topic: {topic} \n Query: {query} \n Related wikipedia knowledge: 
 Topic: {topic} \n Generate a background document from Wikipedia to answer the given question. \n {query} \n 
 Topic: {topic} \n Generate a Wikipedia knowledge to answer the given question.\n Question: {query} \n Wikipedia knowledge: 
 Topic: {topic} \n Generate a Wikipedia to answer the given question.\n Question: {query} \n Wikipedia: | ✓ |
| WoW | Topic: {topic} \n Query: {utterance} \n Related Wikipedia knowledge: 
 Topic: {topic} \n Generate a background document from Wikipedia to reply to the utterance. \n {utterance} \n 
 Topic: {topic} \n Generate a Wikipedia knowledge to answer the given question.\n Utterance: {utterance} \n Wikipedia knowledge: 
 Topic: {topic} \n Generate a Wikipedia to answer the given question.\n Question: {utterance} \n Wikipedia: | ✓ |

Table 11: List of human prompts we tried for zero-shot knowledge generation, evaluated on the validation set of NQ, WoW. {} represents placeholder, and 'utterance' denotes the last utterance of the dialogue partner. We use ✓ to denote the prompt achieving the best performance.

| Dataset | Prompts | Best |
|---------|---------|------|
| NQ | Topic: {topic} \n Query: {query} \n Related Wikipedia knowledge: {knowledge} 
 Topic: {topic} \n Query: {query} \n Knowledge: {knowledge} 
 Topic: {topic} \n Query: {query} \n Document: {knowledge} 
 Topic: {topic} \n Generate a background document from Wikipedia to answer the given question. \n {query} \n {knowledge} 
 Topic: {topic} \n Generate a Wikipedia to answer the given question.\n Question: {query} \n Wikipedia: {knowledge} | ✓ |
| WoW | Topic: {topic} \n Query: {utterance} \n Related Wikipedia knowledge: {knowledge} 
 Topic: {topic} \n Query: {utterance} \n Knowledge: {knowledge} 
 Topic: {topic} \n Query: {utterance} \n Document: {knowledge} 
 Topic: {topic} \n Generate a background document from Wikipedia to reply to the utterance. \n {utterance} \n {knowledge} 
 Topic: {topic} \n Generate a Wikipedia to answer the given question.\n Question: {utterance} \n Wikipedia: {knowledge} | ✓ |

Table 12: List of example templates we tried for few-shot knowledge generation.

| Dataset | Prompts | Best |
|---------|---------|------|
| NQ | Topic: {topic} \n Passage: {knowledge} \n Query: {query} \n Answer: {answer} 
 Topic: {topic} \n Read the passage and answer the question below:\n Passage: {knowledge} \n Question: {query} \n Answer: {answer} 
 Topic: {topic} \n Using the knowledge from the passage to answer the question below:\n Passage: {knowledge} \n Question: {query} \n Answer: {answer} | ✓ |
| WoW | Topic: {topic} \n Passage: {knowledge} \n Speaker 1: {utterance} \n Speaker 2: {response} 
 Topic: {topic} \n Knowledge: {knowledge} \n Speaker 1: {utterance} \n Speaker 2: {response} 
 Topic: {topic} \n Grounding document: {knowledge} \n Speaker 1: {utterance} \n Speaker 2: {response} 
 Passage: {knowledge} \n Query: {utterance} \n Answer: {response} 
 Topic: {topic} \n Using the knowledge from the passage, complete the dialogue below: {knowledge} \n Speaker 1: {utterance} \n Speaker 2: {response} | ✓ |

Table 13: List of example templates we tried for few-shot answer generation.

## B   Analysis of Reference Knowledge

We investigated the importance of reference knowledge in evaluating the factuality of generated knowledge. Specifically, we conducted FLAN-T5 experiments on the WoW dataset using a zero-shot approach. Two sets of experiments were performed: one included reference knowledge in the retrieved evidence pool, while the other did not. Figure 4 illustrates our findings, indicating that the group with reference knowledge exhibits a clear advantage when the number of retrieved evidence is limited. However, as the number of retrieved evidence increases, the performance of both groups converges. These results suggest that reference knowledge is dispensable, particularly when a significant amount of evidence is available. When the number of retrieved evidence surpasses ten, the impact of reference knowledge becomes negligible. We hope this will provide valuable insights for future designs of factuality assessment for generated knowledge.

## C   Details of Baselines

**DPR** (Karpukhin et al., 2020) is a supervised dense retrieval model trained on several QA datasets (including NQ) to retrieve the most relevant Wikipedia passages given a query.

**FLAN-T5** (Wei et al., 2022) is an enhanced version of T5 (Raffel et al., 2020) that is instruction-finetuned in 1.8k NLP datasets to acquire the generalization ability to unseen tasks.

**LLaMA** (Touvron et al., 2023) is an open-source foundation language model trained on publicly available datasets and shows competitive performance with the best models, including GPT-3 (175B) and PaLM-540B.

**ChatGPT** is a sibling model to InstructGPT (Ouyang et al., 2022) that is trained to follow instructions in a prompt and provide a detailed response. We adopt text-davinci-003 version for

evaluation.

## D  Details of Datasets

**Natural Questions (NQ)** (Kwiatkowski et al., 2019) is an open-domain QA dataset, where the questions are mined from real Google search queries. The corresponding ground truth knowledge and the answers to the questions are paragraphs and short spans in the Wikipedia pages.

**Wizard of Wikipedia (WoW)** (Dinan et al., 2018) is a knowledge-grounded dialogue dataset designed for information-seeking scenarios, where one speaker introduces knowledge related to a topic to the other speaker by grounding his/her responses in a specific sentence from a Wikipedia page.

## E  Implementation Details

All the metrics we designed are model-based metrics, utilizing solely off-the-shelf models. We present the models used in Table 14.

## F  Prompts for Knowledge Generation

### F.1  Zero-shot Prompts

In our experiments, we observed that zero-shot prompting was highly unstable. Therefore, we conduct experiments using multiple human prompts and select the most effective ones for the WoW and NQ datasets. The human prompts we evaluate are listed in Table 11.

### F.2  Few-shot Prompts

In the few-shot setting, our prompt is constructed using k randomly chosen examples from the training set:

$$prompt = (example_1 \text{\textbackslash n} ... example_k \text{\textbackslash n} example_{test})$$

The example templates utilized for knowledge generation are provided in Table 12. Please note that $example_{test}$ differs from $example_i$ as it does not contain $\{knowledge\}$ in the placeholder.

## G  Prompts for Answer Generation

We adopt few-shot prompting on the LLaMA model in answer generation and the example templates used for answer generation are provided in Table 13.

## H  Detailed Correlations between Intrinsic and Extrinsic Metrics

We listed the detailed correlations between intrinsic and extrinsic metrics for LLaMA, FLAN-T5, and ChatGPT on the NQ dataset in Table 15.

## I  Table of Model Size Impact

We list the specific numerical values of performance scaling with the model size in Table 16, including LLaMA-65B/33B/7B and FLAN-T5-11B/3B/780M.

## J  Details of Human Evaluation

### J.1  Human Annotation

We conducted a human evaluation with 400 samples from the NQ and WoW test set. Among these, 320 samples were from the zero-shot setting in the NQ dataset, involving all four models, while 80 samples were from the few-shot setting in the WoW dataset, involving one model (ChatGPT). Three expert annotators, who were familiar with the tasks, were employed to rate the acquired knowledge and generated answers based on four intrinsic perspectives and two extrinsic perspectives. Each perspective was scored on a scale of 0, 1, or 2, representing unacceptable, acceptable, and excellent, respectively. The average kappa value of the annotation is 0.612 on 20% cross-annotation data. The detailed instructions for the human annotation can be found in Table 17.

Note for factuality assessment, the reliable evidence for the generated knowledge $k$ is acquired by the following process: For each sentence in $k$, we use it as the query to search Google, and regard the top1 Wikipedia webpage as a reliable knowledge source to verify the factuality of this sentence. Another point worth noting is that for the evaluation of validity in WoW, we reused the factuality evaluation process since the responses in WoW are open-ended.

### J.2  Human Evaluation Results on WoW

Based on the provided annotations, we assessed the correlation between ChatGPT's automatic metrics and human judgement on the WoW dataset. The results are presented in Table 18.

### J.3  Baseline Metrics

We compared it with three reference-reliant metrics in knowledge evaluation. Their definitions and calculation methods are as follows:

| Metric | Model | Link |
|---|---|---|
| Factuality | NLI-RoBERTa-large ColBERTv2 | https://huggingface.co/sentence-transformers/nli-roberta-large https://github.com/stanford-futuredata/ColBERT |
| Relevance | BERT-ranking-large | https://github.com/nyu-dl/dl4marco-bert |
| Coherence | GPT-neo-2.7B Coherence-Momentum | https://huggingface.co/EleutherAI/gpt-neo-2.7B https://huggingface.co/aisingapore/coherence-momentum |
| Informativeness | GPT-neo-2.7B | https://huggingface.co/EleutherAI/gpt-neo-2.7B |
| Helpfulness | LLaMA-65B | https://github.com/facebookresearch/llama/tree/main |
| Validity | NLI-RoBERTa-large ColBERTv2 | https://huggingface.co/sentence-transformers/nli-roberta-large https://github.com/stanford-futuredata/ColBERT |

Table 14: List of all models that we use in designing our framework.

| Model | Extrinsic | Instrinsic | | | | |
|---|---|---|---|---|---|---|
| | | Fact. | Rel. | Coh-sent. | Coh-para. | Info. |
| FLAN-T5 | helpful. | $0.15^{\dagger}$ | $-0.21^{\dagger}$ | $0.20^{\dagger}$ | $-0.21^{\dagger}$ | 0.02 |
| | validity | $0.23^{\dagger}$ | $-0.16^{\dagger}$ | $0.14^{\dagger}$ | $-0.10^{\dagger}$ | 0.07 |
| LLaMA | helpful. | 0.03 | 0.05 | 0.06 | $-0.09^{\dagger}$ | -0.01 |
| | validity | $0.09^{\dagger}$ | 0.07 | 0.05 | -0.06 | -0.03 |
| ChatGPT | helpful. | $0.16^{\dagger}$ | 0.03 | 0.08 | 0.02 | $-0.04^{\dagger}$ |
| | validity | $0.22^{\dagger}$ | $0.13^{\dagger}$ | $0.02^{\dagger}$ | $0.09^{\dagger}$ | 0.03 |

Table 15: The Somers' correlation between intrinsic and extrinsic metrics in zero-shot setting on NQ. Correlation scores with $p\text{-}value < 0.05$ are marked with $^{\dagger}$.

| Model | Size | Fact. | Rel. | Coh. | Info | Help. | Val. |
|---|---|---|---|---|---|---|---|
| **LLaMA** | 65B | **0.942** | **0.732** | **0.824** | **0.757** | **0.219** | **0.420** |
| | 33B | 0.656 | 0.633 | 0.734 | 0.608 | 0.203 | 0.402 |
| | 7B | 0.773 | 0.626 | 0.805 | 0.662 | 0.154 | 0.375 |
| **FLAN-T5** | 11B | 0.584 | 0.685 | 0.778 | 0.673 | **-0.146** | **0.325** |
| | 3B | **0.657** | 0.663 | **0.816** | 0.708 | -0.155 | 0.324 |
| | 780M | 0.506 | **0.729** | 0.793 | **0.729** | -0.162 | 0.252 |

Table 16: Performance on NQ with varying sizes of FLAN-T5 and LLaMA as knowledge generators. The max(0, .) operation in Eq.6 has been excluded to emphasize the sequential relationship among different sizes of FLAN-T5. **Bold** and Underlined results represent the best and second-best performances for each model, respectively.

**Hallucinated NE Ratio (HE)** (Lee et al., 2023) proposed a NE-based metric that is designed with an intuition that a model is hallucinating (making factual errors) if it generates an NE that does not appear in the reference knowledge source. The NE-based metric can be calculated as: $\text{HNE} = |\text{HALLU}_{\text{NE}}| / |\text{ALL}_{\text{NE}}|$ where $\text{ALL}_{\text{NE}}$ is the set of all the NEs detected in the LM generation, and $\text{HALLU}_{\text{NE}}$ is a subset of $\text{NE}_{\text{All}}$ that does not appear in the reference Wikipedia knowledge. Note that evaluating $\text{NE}_{\text{ER}}$ requires the existence of refer-

ence knowledge. We adopt $-\text{HNE}$ when computing the correlation with human judgement.

**Entailment Ratio (ER)** (Lee et al., 2023) also introduces an NLI-based approach to assess factual knowledge by measuring its entailing relationship with ground-truth/reference knowledge. The entailment ratio is computed as follows: $\text{Entail}_{\text{R}} = |\text{ENTAIL}_{\text{gen}}| / |\text{ALL}_{\text{gen}}|$, where $\text{ALL}_{\text{gen}}$ is a set of all generated knowledge, and $\text{ENTAIL}_{\text{gen}}$ is the set of generated knowledge that can be entailed by the NLI model. Specifically, we use the entailment probability of each example as its ER score.

**F1 of knowledge (F1)** (Liu et al., 2022) employs a unigram F1 score to evaluate the quality of generated knowledge. This metric measures the overlap between the generated knowledge and the reference knowledge by evaluating word-level matches. By assessing the degree of agreement, the F1 metric provides an estimation of the knowledge quality, specifically from a relevance perspective.

**NLI-weak-supervised** (Kryscinski et al., 2020b) train a classification model on constructed data to perform consistency checking on (document, sentence) pairs. We chose the factCC version as our baseline.

**NLI-decompose-claim** (Glover et al., 2022b) found that in general, sentence-level decomposition is preferable for the hypothesis side of the NLI input. So we also decompose the generated knowledge into sentences and then aggregate the sentence-level scores to produce a document-level score.

**NLI-multitask** fine-tunes the DeBERTa-v3-large model on FEVER and two NLI datasets.

**Exact Match (EM)** (Rajpurkar et al., 2016) use Exact Match to measure the percentage of predictions that match its ground truth answers exactly.

| Dimension | Value | Description |
|---|---|---|
| Factuality | 2 | All sentences in $k$ are factually accurate and the information in them can be verified with reliable evidence. |
|  | 1 | $k$ contains at least one sentence with non-verified information, while others are factually accurate. |
|  | 0 | $k$ contains at least one sentence with at least one factual error that is inconsistent with reliable knowledge. |
| Relevance | 2 | $k$ is highly relevant to the topic and query/utterance. |
|  | 1 | $k$ is relevant to the topic but less relevant to the query/utterance. |
|  | 0 | $k$ is irrelevant to both the topic and query/utterance. |
| Coherence | 2 | $k$ is very coherent and fluent (do not consider the truncation at the end due to the maximum generation length). |
|  | 1 | $k$ has some minor incoherence or lack of fluency, *e.g.*, phrase or sentence repetition, but it does not affect understanding. |
|  | 0 | $k$ has significant coherence and fluency issues that are hard to understand. |
| Informative | 2 | $k$ contains informative content that you don't know before. |
|  | 1 | $k$ contains limited or trivial information against your knowledge. |
|  | 0 | $k$ fails to provide any meaningful information. |
| Helpfulness | 2 | $k$ directly provides or contains the correct answer. |
|  | 1 | $k$ indirectly help in generating the correct answer. |
|  | 0 | $k$ does not contain any useful information for the correct answer. |
| Validity | 2 | The answer generated based on $k$ is correct. |
|  | 1 | The correctness of the generated answer cannot be determined. |
|  | 0 | The answer generated based on $k$ is completely incorrect. |

Table 17: Annotation guideline of LLM generated knowledge.

| Model | Fact. | Rel. | Coh. | Info. | Help. | Val. |
|---|---|---|---|---|---|---|
| ChatGPT | 0.71 | 0.57 | 0.52 | 0.40 | 0.79 | 0.54 |

Table 18: Somer's D correlation of metrics with the human annotation on WoW ). $p\text{-}value$ for all results are $< 0.05$. We report the maximum for coherence.