# OpenReview forum: "Beyond Factuality: A Comprehensive Evaluation of Large Language Models as Knowledge Generators"
_EMNLP/2023/Conference — EMNLP 2023 Main_

### Official Review · Reviewer_QUWs · 2023-08-03

**Typos Grammar Style And Presentation Improvements:** Line 296
**Soundness:** 4

**Excitement:**

4: Strong: This paper deepens the understanding of some phenomenon or lowers the barriers to an existing research direction.

**Paper Topic And Main Contributions:**

This paper develops a comprehensive knowledge evaluation framework (CONNER) to evaluate the LLM's knowledge generation capability. CONNER involves both intrinsic and extrinsic evaluation, where the intrinsic assessment focuses on the quality of generated knowledge while the extrinsic aspect considers the applicability of the generated knowledge to specific downstream tasks (QA, knowledge-grounded dialogue). The evaluation results contribute to the understanding of how model generated knowledge affects downstream tasks, and yields practical experience for how to leverage LLM-generated knowledge for downstream tasks.

**Questions For The Authors:**

1. In the calculation of the Informativeness score, it is unclear how Eq.(5) is related to Eq.(6). In my understanding, the $k$ in $H(k|q)$ on the LHS of Eq.(5) is a random variable rather than a $k$ with a definite value. $H(k|q)$ here represents the entropy of the knowledge distribution under the given query $q$, with a larger entropy implying a more uniform knowledge distribution under the given query. But the final Informativeness score in Eq.(6) is solely obtained by subtracting the geometric mean of the generation probabilities of all tokens in $k$ from 1 under the given $q$ and $k$, which seems unrelated to the entropy defined in Eq.(5). Could you explain the connection here?
2. In the calculation of the Helpfulness score, is there a coefficient $u$ missing in the numerator of Eq.(7)? Also, there lacks a necessary explanation of the relationship between $L(q, k, a)$ and $P(a|q, k)$.

**Reasons To Accept:**

1. This paper is the first to propose an automatic evaluation framework for LLM-generated knowledge. Many previous works have utilized the knowledge generated by LLMs, but lack an evaluation of whether this knowledge is reliable or effective. This work fills this gap nicely.
2. The experimental part of the paper investigates the quality of the knowledge generated by the model and its impact on downstream tasks through Somers’ correlation score, which provides guidance for how to better use model-generated knowledge in downstream tasks.
3. The authors conduct a sufficient number of manual experiments and show a high correlation with the scores automatically obtained by CONNER, suggesting the reliability of this method for automatic evaluation.

**Reasons To Reject:**

1. Some of the metrics used by this method are not explained clearly enough. (Refer to question 1 and 2)

**Reproducibility:**

4: Could mostly reproduce the results, but there may be some variation because of sample variance or minor variations in their interpretation of the protocol or method.

**Reviewer Confidence:**

4: Quite sure. I tried to check the important points carefully. It's unlikely, though conceivable, that I missed something that should affect my ratings.

---

> ### Author Rebuttal · Authors · 2023-08-29
>
> We are profoundly grateful for your positive and insightful feedback on our work! Your recognition of our paper being the pioneer to propose an automatic evaluation framework for LLM-generated knowledge, and its role in filling a gap in existing literature is highly encouraging. Your endorsement of our experimental investigations' guidance for downstream tasks, and the reliability through a sufficient number of manual experiments are particularly encouraging. We will now address your points individually.
>
>
> #### **Response for Informativeness**
> We appreciate your insightful feedback. We realize that there appears to be a disconnect between Eq.(5) and Eq.(6). As you noted, Eq.(5)'s $H(k|q)$ represents the entropy of the knowledge distribution for a specific query, with a lower value implying a more nonuniform knowledge distribution under the given query.
>
> However, computing $H(k|q)$ requires a distribution over an infinite set of variable-length knowledge sequences, which is impractical as we only possess information for a specific 'k'. To address this, we employ an autoregressive language model to factorize the joint distribution over 'k' into a product of conditionals over a finite set of tokens as follows:
>
> $P(k | q)= \\prod\\nolimits\_{t=1}^{M} P( k_t | k\_{1:t-1}, q)$
>
> This conditional independence assumption permits us to define a model on a finite token space of knowledge sequence. Therefore, instead of the sequence level, we estimate uncertainty at the token level, which considers uncertainty in the prediction of a single 'k'.
>
> We then employ an approximation method, as suggested by [1][2] and apply '1 - the geometric mean of the generation probabilities of all tokens in k under the given q'. This serves as an effective approximation of uncertainty reduction, which we denote as $S\_{info}$. Empirically, it aligns with $H(k|q)$ in that a score of 0 indicates the absence of useful information, which corresponds to a case where $P(k|q)=1$, suggesting the most uneven knowledge distribution.
>
> In summary, Eq.(5) represents an ideal estimation of knowledge distribution, while Eq.(6) provides a practical approximation under specific knowledge.
>
> We hope this explanation clarifies the relationship between these two equations. Your astute observation has helped us to provide this clarification.
>
> &nbsp;
>
> #### **Response for Factuality Equation**
> Apologies for the confusion.  The clearer representation of the equation should be as follows:
>
> $
> S_\texttt{help}(q, a, k, k_{1}^-, \cdots, k_{u}^-)
> =\max \left(0, 1 - \frac{\mathcal{L}(q,k,a)}{\frac{1}{u} \sum\nolimits_{i=1}^{u} \mathcal{L}(q,k_{i}^-,a)}\right)
> =\max \left(0, 1 - \frac{\log P(a|q, k)}{\frac{1}{u} \sum\nolimits_{i=1}^{u} \log P(a|q,k_{i}^-)}\right)
> $
>
> Here, $\mathcal{L}(q,k,a)$ signifies the cross-entropy loss of the answer generation process, and $P(a|q,k)$ denotes the probability of the correct answer. We calculate the cross-entropy loss based on this probability.
>
> We will ensure that this correction, as well as the explanations, are incorporated in our revised version.
>
> &nbsp;
>
> #### **Response for Typo**
> Thank you very much for pointing out the typo. We will correct this in our revised manuscript. Your attention to detail is greatly appreciated.
>
> &nbsp;
>
> **References**
>
> [1] Kenton Murray et. al 2018, Correcting length bias in neural machine translation
>
> [2] Andrey Malinin et. al 2021, Uncertainty estimation in autoregressive structured prediction

---

### Official Review · Reviewer_2CNk · 2023-08-05

**Typos Grammar Style And Presentation Improvements:** 1. Since the goal of the paper is an …
**Soundness:** 4

**Excitement:**

4: Strong: This paper deepens the understanding of some phenomenon or lowers the barriers to an existing research direction.

**Missing References:**

[1] Glover et. al 2022, Revisiting text decomposition methods for NLI-based factuality scoring of summaries

[2] Utama et. al 2022, Falsesum: Generating Document-level NLI Examples for Recognizing Factual Inconsistency in Summarization

[3] Honovich et. al 2022 TRUE: Re-evaluating Factual Consistency Evaluation

[4] Ji et, al. 2022, Survey of Hallucination in Natural Language Generation

[5] Schuster et, al 2021, Retrieval augmentation reduces hallucination in conversation

[6] Komeili et al 2021, Internet-Augmented Dialogue Generation

[7] Chen et al 2017, Reading Wikipedia to Answer Open-Domain Questions

[8] Pagnoni et al 2021, Understanding Factuality in Abstractive Summarization with FRANK: A Benchmark for Factuality Metrics

[9] Devaraj et al 2022, Evaluating Factuality in Text Simplification

[10] Liu et, al 2020, Evaluating Text Coherence at Sentence and Paragraph Levels

**Paper Topic And Main Contributions:**

The paper aims to evaluate LMs as knowledge generators. Recent works have proposed to leverage LMs to generate text to be used as context for knowledge-intensive tasks. But there is a concern about the quality and factual inaccuracies in the generated text and the impact of such inaccuracies in the downstream task. This paper addresses such concerns by proposing to compresively evaluate such generated knowledge across six factors — Factuality, Relevance, Coherence, Informativeness, Helpfulness and Validity. They propose an automatic metric for each aspect and evaluate generated knowledge from 3 LLMs (Flan-T5, Llama and ChatGPT) and text from retrieval system (DPR) across using these metrics. Their results indicate that LLM generate text are less factual but more relevant than retrieved text. The lower factuality doesn’t significantly effect the downstream tasks since the LLM generated text have higher helpfulness and validity. The additionally experiment and show effect of model size, long tail generation and long form generation on factuality of the generated text. Finally they show how to leverage these metrics to improve the few-shot prompts and generated text to get better performance on the downstream tasks.
The main contributions of this work:

1. A framework for evaluating knowledge generated from LMs for knowledge intensive task.

2. Automatic metrics for each aspect of generation to be evaluated.

3. Evaluation of model generated text in context of knowledge intensive tasks like open-domain QA and knowledge grounded dialogue.

**Questions For The Authors:**

A. For the ranking BERT model, which model from the relevant paper is used. The paper proposed two variants for ranking, which one was used and how was it adapted for measuring relevance?

B. Line 269 typo: coh_para should be coh_sent

C. Line 298 typo - S_ent ?

D. For the informativeness measure entropy could be increased without the text being informative? Any text as long as it’s new will be considered informative. Is that what the expected behaviour of the metric ?

E. Line 322 - clarify where the Loss is being used exactly.

F. Table 2 - why is sentence coherence so low for dpr (basically all models) ? What does it mean if coherence values are so low ?

G. For qualitative examples (Table 4), please include paragraphs for the same query for better comparison (dpr query is different).

H. Para from line 407 - how was this correlation done? What does correlation between metrics actually reveal - more clarification on what this experiment actually measures would be good.

I. Line 550 type 43.45

**Reasons To Accept:**

1. First evaluation targeted towards understanding the use of model generated text as context for knowledge intensive tasks. The paper highlights the importance of studying the context and its impact in addition to just measuring downstream performance.
2. Interesting findings - With a lot of concerns about factuality of generated text, the findings that even with lower factuality, the impact on downstream task impact is not significant will be interesting to the community. They show that relevance and coherence of the outputs are more important that small factual mistakes.

**Reasons To Reject:**

1. Motivation and explanations for each metric proposed is not clear - The evaluation of some aspects do not follow standard evaluation measures and some new automatic metrics are used without providing sufficient motivation. There is minimal grounding in prior research in such evaluation metrics and does not discuss limitations in them to warrant new metrics. Since the goal of the paper is to evaluate models, the metrics used to measure quality of generation should be robust and accurate to ensure that findings are reliable. Therefore motivation and explanations on why the existing metrics were not applicable is important. I am not fully convinced that the current evaluation measures chosen adequately measure the concerned aspect, which cast doubt on the validity of the findings.

    i) Factuality - Prior work has explored evaluating factuality of text given evidence [1,4, 5] via models trained to predict factuality. [2,3] has shown limitation of using off-the-shelf NLI models for detecting factual errors. How does the proposed metric connect to previous evaluation measures and is it experimentally better at detecting factual errors?

    ii) Relevance - Following [6, 7], F1 and Knowledge-F1 or Accuracy wrt to retrieved/gold evidence is commonly used to measure relevance. It’s not clear how a BERT ranking model would measure relevance well. Explaining the motivation of using a BERT ranking model and citing and comparing with the prior evaluation measures is needed.

    iii) Extrinsic Evaluation - Prior work Open-Domain QA use Exact-Match and F1 scores  for evaluating downstream task performance [8,9]. The helpfulness metric measured using ‘cross entropy losses of answer generation’ is interesting but approximate and not exact metrics. Validity metric evaluates entailment which again is a semantic match but might not compute exact evaluation of the downstream performance. Incorporating the prior metrics in addition to the proposed ones or comparing against them to show the effectiveness of the new metric is important.

2. Weak evaluation of proposed metrics - Since some of the metrics proposed are not commonly used or are new, the first evaluation should center around the efficacy of the evaluation measures. Currently, the paper focuses on findings before establishing the efficacy of metrics in measuring different aspects. The correlation study done in Sec 6 is limited and very briefly explained.

    i) Effect on model and dataset bias - [10] discusses the concern of clubbing data from multiple models and datasets for computing correlations. Biases like ‘model detects one dataset/model well’ leak into the correlation and compromise final results. The paper suggests computing and reporting ‘partial correlation’ to correct this problem.

    ii) Human and metric are measuring different aspects and hence not comparable - The metrics proposed and used in the paper are ‘Factuality, Relevance, Coherence, Informativeness, Helpfulness and Validity’ and the human annotations obtained are a 0,1,2 label for ‘Unacceptable, Acceptable, and Excellent’. I don’t think these are comparable, especially considering that each correlation for each metric is computed against the same scores. Ideally, most work obtain human annotations for each aspect being measured and compute correlations for each pair.

    iii) Using Sommers D for correlation - What is the motivation for using Sommers D? The statistic measures the association between an ordinal dependent variable and an ordinal independent variable. How is it applicable in this setting. Most prior work use Pearson correlation or Spearman rank correlation [10, 11, 12], is there a reason a different statistic was chosen ?

3. Weak baselines in metric evaluation - As mentioned in motivation for metrics, prior research in different evaluation measures need to be citied and compared against. E.g, [1,2] should be used as factuality baselines, [6,7] should be used for relevance in addition to F1.

[1] Kryściński et. al 2019, Evaluating the Factual Consistency of Abstractive Text Summarization

[2] Glover et. al 2022, Revisiting text decomposition methods for NLI-based factuality scoring of summaries

[3] Utama et. al 2022, Falsesum: Generating Document-level NLI Examples for Recognizing Factual Inconsistency in Summarization

[4] Honovich et. al 2022 TRUE: Re-evaluating Factual Consistency Evaluation

[5] Ji et, al. 2022, Survey of Hallucination in Natural Language Generation

[6] Schuster et, al 2021, Retrieval augmentation reduces hallucination in conversation

[7] Komeili et al 2021, Internet-Augmented Dialogue Generation

[8] Chen et al 2017, Reading Wikipedia to Answer Open-Domain Questions

[9] Karpukhin et al 2020, Dense Passage Retrieval for Open-Domain Question Answering

[10] Pagnoni et al 2021, Understanding Factuality in Abstractive Summarization with FRANK: A Benchmark
for Factuality Metrics

[11] Devaraj et al 2022, Evaluating Factuality in Text Simplification

[12] Dror et al 2018, The Hitchhiker’s Guide to Testing Statistical Significance in Natural Language Processing

**Reproducibility:**

4: Could mostly reproduce the results, but there may be some variation because of sample variance or minor variations in their interpretation of the protocol or method.

**Reviewer Confidence:**

4: Quite sure. I tried to check the important points carefully. It's unlikely, though conceivable, that I missed something that should affect my ratings.

---

> ### Author Rebuttal · Authors · 2023-08-29
>
> We would like to thank the Reviewer 2CNk for the detailed comments and careful reading. Your recognition of our work as the first evaluation focused on understanding the use of model-generated knowledge is inspiring. We are also heartened that you find our findings interesting and valuable to the community. Following this, we will sequentially address your comments, provide clarifications on some issues, and enhance our work with new experimental results.
>
> ### **Response for Motivation**
> To address the concern about our motivation, we first elaborate on the limitations of existing metrics for each perspective and then discuss the motivation and design process of our framework.
>
> **1. Limitations of existing metrics.** We have supplemented the motivation for the Factuality, Relevance, and Extrinsic metrics mentioned in your review due to their unique design compared to existing work.
>
> ● **Factuality.** Existing factuality metrics [1,2,4] normally determine the factual correctness of generated content by assessing their consistency with golden/human-annotated references. This method proves effective for summarization tasks but exhibits significant shortcomings in the burgeoning field of knowledge generation.
>
> One major limitation is the potential discrepancy between 'ground-truth knowledge' in the dataset and model-generated content. For instance, a model may generate knowledge beyond the reference knowledge scope. The correctness of these extra portions cannot be verified against reference knowledge, potentially leading to the low correlation between existing factuality scores and human annotations, as shown in **Table 1**.
>
> Furthermore, the absence of annotated references in real-world situations renders these metrics ineffective. This highlights the urgent need for more adaptive factuality evaluation methods in the evolving domain of knowledge generation.
>
>
> ● **Relevance.** Previous relevance metrics usually rely on word overlap or similarity with human-written references [6,7]. However, in the context of knowledge generation, these metrics encounter two significant drawbacks:
>
> First, They struggle to align well with the knowledge acquisition scenario, where large language models (LLMs) act as generative search engines. The main goal here is to assess the relevance of the generated knowledge to the given query, similar to a search engine's role. In this context, measuring similarity with annotated knowledge is indirect and may not accurately represent relevance, potentially leading to the low correlation between these metrics and human-annotated relevance scores, as depicted in **Table 10** in the paper.
>
> Second, similar to factuality metrics, reference reliance becomes a significant bottleneck in real-world applications where such references are not readily available.
>
> ● **Extrinsic.** Extrinsic metrics such as EM (Exact Match) and F1 score can indeed be used to assess the quality of answers. However, their over-reliance on literal matches often results in low recall. For instance, "PRC" and "China" refer to the same entity, but these metrics would yield a score of 0 due to the difference in literal representation. This limitation is likely the reason for their underperform in **Table 2**.
>
> Additionally, their assessment of "quality" is rather vague. For example, we may wish to gauge the factual accuracy of downstream tasks using the generated knowledge or assess how much the upstream knowledge aids downstream tasks. These prompt the need for more granular downstream metrics.
>
> **2. Motivation and design process - “Why did we choose these six metrics”.**
> Given these limitations, there is a clear need for a framework that can provide a fine-grained evaluation of knowledge generation from essential perspectives in reference-free scenarios. This was the primary motivation for designing our proposed framework.
> The process of identifying these essential perspectives began with a detailed manual investigation of the LLaMA's responses to 160 samples from the NQ and WoW datasets (80 each). We sought to understand the errors in knowledge generation and answer generation made by the model.
>
> To achieve this, we employed thematic analysis [A1], a widely recognized method for identifying patterns within data. Our process started with the extraction of preliminary patterns from subtle errors in the knowledge and answers, which were then aggregated into broader patterns. We further validated these patterns using an additional 40 questions, ensuring that they adequately represented the spectrum of errors.
>
> As a result, we discerned four primary error categories in knowledge generation and two in answer generation. These categories formed the foundation for our four intrinsic metrics for knowledge evaluation and two extrinsic metrics for answer evaluation.
>
> We hope these explanations help improve the understanding of our work to warrant new metrics.
>
> &nbsp;
>
> ### **Response for Factuality**
>
> Thank you for your insightful comments. We have organized our response to more clearly address the three key areas you've mentioned: the connection with previous work, comparisons with advanced models, and why we choose off-the-shelf models.
>
> **"Connection to Previous Evaluation Measures".**  Our work significantly diverges from previous studies[1,2], especially in defining and evaluating 'factuality'. The latter focuses on 'closed' tasks like abstractive summarization, dealing with input-output contradictions. As noted in **lines 157-166**, they evaluate 'factuality' by comparing input and output, requiring a more advanced, task-specific NLI model.
> Conversely, our open-ended tasks like open-domain QA and dialogue do not permit direct correctness judgement based on the input question. 'Factuality' is assessed by how well the model's output aligns with world knowledge, necessitating the **output decomposition**, **detailed evidence collection** and **results aggregation** processes as integral parts of our factuality metrics.
>
> **Compared with More Advanced Baselines.** As shown in Table 1, we have also supplemented three more advanced methods as the baselines of our Factuality metrics and computed their correlation with human annotation of actuality.
>
> (1)  NLI-multitask, which fine-tunes the DeBERTa-v3-large model on FEVER and two NLI datasets.
>
> (2)  weak-supervised[1], which was trained on constructed data to perform consistency checking on (document, sentence) pairs. We chose the factCC version as our baseline.
>
> (3)  NLI-decompose-claim[2], which found that in general, sentence-level decomposition is preferable for the hypothesis side of the NLI input. So we decompose the generated knowledge into sentences and then aggregate the sentence-level scores to produce a document-level score.
>
> |Metric|DPR|FLAN-T5|LLaMA|ChatGPT|
> |:---:|:---:|:---:|:---:|:---:|
> |Factuality|0.65|0.66|0.66|0.63|
> |NLI-off-the-shelf|0.22|0.6|0.22|0.17|
> |NLI-multitask|0.18|0.51|0.26|0.32|
> |weak-supervised[1]|0.12|0.43|0.19|0.16|
> |NLI-decompose-claim[2]|0.23|0.47|0.27|0.38|
> |
>
> Table 1. Correlation of Factuality Scores with Human Evaluation Across Different Baseline Methods
>
> Our approach delivers competitive Factuality scores across all models. This is because our method decomposes the generated knowledge into individual sentences, and collects corresponding evidence for each sentence. This allows for a more comprehensive coverage of every component of the generated knowledge, as opposed to simply using ground-truth knowledge. This nuanced approach enhances the model's capacity to accurately generate factual information, resulting in an improved Factuality score.
>
> **Why do we "use off-the-shelf model".** While fine-tuning off-the-shelf NLI models for specific tasks can help overcome limitations, our goal is to provide a more universally applicable solution. Our preference for general models over task-specific ones ensures that our evaluation framework remains useful across a broad range of knowledge-intensive tasks, even in real-world scenarios.
>
> We appreciate your keen insights and will ensure these points are clarified in the revised manuscript.
>
> &nbsp;
>
> ### **Clarification for Relevance**
>
> We divide our response into two parts to respond to your questions about the motivation for using a BERT ranking model and to make some clarification about baselines.
>
> **1. Motivation for Using BERT Ranking Model.**
> Our choice to use a BERT ranking model is driven by three main factors:
>
>  **(1) Knowledge Generation as a Generative Search Process.** In the process of generating knowledge, large language models (LLMs) effectively function as generative search engines. The most crucial aspect of such a process is to evaluate the relevance of the acquired knowledge document to the given query. The BERT ranking model's training data comes from the Bing search engine, with results manually annotated for relevance. This makes the BERT ranking model particularly suited to precisely evaluate the relevance of a document (or in our case, knowledge generated by a language model) to a given query.
>
> **(2)  Absence of Reference in Real-World Scenarios.** In real-world applications, ground-truth knowledge is often not available. The BERT model, trained to rank matching pairs of queries and knowledge documents, proves effective in estimating the relevance of the knowledge to the given query, even in the absence of golden knowledge.
>
> **(3) Potential Misalignment with Pre-Annotated Reference.** Even when datasets have pre-annotated references, they may not align perfectly with the knowledge generated by the model. In our dataset, the 'ground-truth knowledge' often corresponds to annotations made against pre-existing human answers, which may not perfectly align with the knowledge generated by the model. Therefore, we find it more meaningful to measure the retrieval relevance between the query and the model-generated knowledge.
>
>
> **2.  Clarification on Baselines**
> We understand your concerns about the baselines of relevance evaluation and wish to offer some clarifications:
>
> ● **F1 Score:** The F1 score was indeed calculated in our evaluation. As outlined in Table 10 of our paper, we used the F1 score to measure the overlap between the ground-truth knowledge and the knowledge generated by the model.
>
> ● **Knowledge-F1**: Knowledge-F1 is not designed for evaluating knowledge generation. It measures the overlap between the model's response and the knowledge used during response generation. This makes Knowledge-F1 more suitable for assessing response generation in knowledge-grounded dialogue tasks, not knowledge-generation tasks.
>
> ● **Accuracy**: Accuracy is not suitable for text generation tasks, which is the focus of our work.
>
> We appreciate your insightful feedback. It has helped us clarify our research and will undoubtedly improve our work. Thank you for your thorough review.
>
> &nbsp;
>
> ### **Response for Extrinsic Metrics**
> To address your suggestions, we supplemented our extrinsic evaluation with a comparison of EM and F1 scores and calculated their correlation with human evaluations. Given that EM and F1 scores are primarily used to assess the correctness of downstream answers, we utilized them as baselines for our validity metric.
>
> The results of our supplementary evaluations are in Table 2:
>
> |Metric|DPR|FLAN-T5|LLaMA|ChatGPT|
> |:---:|:---:|:---:|:---:|:---:|
> |Validity|0.83|0.73|0.85|0.82|
> |EM|0.59|0.51|0.54|0.61|
> |F1|0.74|0.67|0.76|0.77|
> |
>
> Table 2. Correlation with Human Annotation of Answer Correctness Across Different Methods
>
> From the results, it is evident that our validity metric demonstrates a significantly higher correlation with human evaluation of answer correctness compared to the traditional metrics.
> We will add this evaluation in our revised version. We hope that these additions address your concerns and illustrate the effectiveness of our extrinsic evaluation.
>
> &nbsp;
>
> ### **Response for "focuses on findings before establishing the efficacy of metrics"**
> We understand your concerns about the presentation order in our paper.  In our initial structure, we chose to present the validation of the metric efficacy in Chapter 6 and Appendix J, intending to first share the exploratory findings and analyses, which we believed might be of more immediate interest and value to readers.
>
> However, we acknowledge your insightful point about the importance of showcasing the metrics' effectiveness earlier in the paper. We appreciate your suggestion and will make adjustments to enhance the rigour of our presentation.
>
> &nbsp;
>
> ### **Clarification for "clubbing data from multiple models and datasets for computing correlations"**
> We would like to clarify that we did not "club data from multiple models and datasets for computing correlations". Rather, we have calculated correlations separately for each dataset and model. As outlined in **Table 10 and Appendix J.1** in our manuscript, each model's performance has been independently evaluated on each dataset, without merging the results across different data sources. By adopting this approach, we have strived to ensure that our analysis is as fair and unbiased as possible, and that it reflects the true performance of each model on each dataset.
>
> We hope this addresses your concern about the correlation study.
>
> &nbsp;
>
> ### **Clarification for "each correlation for each metric is computed against the same scores."**
> We would like to clarify that we did not use a uniform score for correlation computations w.r.t different metrics.  Instead, as mentioned in **lines 556-557 and 957-959** of our manuscript, human annotators assigned scores of 0, 1, 2 for each of the intrinsic and extrinsic metrics we used, representing 'Unacceptable, Acceptable, and Excellent', respectively.
>
> Furthermore, we provided detailed 'Human annotation guidelines for each perspective/metric' in **Table 17** of our paper. This approach ensures that each metric has a direct and meaningful comparison with human judgments.
>
> We hope this clarification addresses your concern about the comparability between human annotations and individual metrics.
>
> &nbsp;
>
> ### **Response for "Use of Somers' D Correlation"**
> We appreciate your attention to the choice of correlation measurement used in our study. We would like to clarify our rationale for selecting Somers' D and provide additional analysis using the Pearson correlation to address your concerns.
>
> **Motivation for Using Somers' D.**
> Our choice of Somers' D correlation was primarily motivated by its superior performance in handling ties of a biased random variable [A2]. In this method, correlations are measured against a biased random variable $X \in[0,1]$, represented by the potential perturbation or error presence indicator within data. For each score Y considered, it is calculated using the formula $D(Y\mid X)=\tau(X, Y) / \tau(X, X)$. This capability of Somers' D is particularly beneficial for our study and aligns with our evaluation characteristics.
>
> **Supplementary Analysis Using Pearson Correlation.**
> To address your concern further, we conducted a supplementary analysis using the Pearson correlation. The results are in Table 3, which mirrors those obtained with Somers' D, strengthening the robustness of our conclusions. We present a comparison of the results obtained using both correlation methods:
>
> |Metric|Correlation|DPR|FLAN-T5|LLaMA|ChatGPT|
> |:---:|:---:|:---:|:---:|:---:|:---:|
> |Factuality|Somers' D|0.65|0.66|0.66|0.57|
> | |Pearson|0.74|0.68|0.67|0.57|
> |Relevance|Somers' D|0.69|0.37|0.55|0.54|
> | |Pearson|0.75|0.4|0.64|0.64|
> |Coherence|Somers' D|0.53|0.58|0.44|0.49|
> | |Pearson|0.48|0.51|0.36|0.42|
> |Informative|Somers' D|0.3|0.17|0.35|0.32|
> | |Pearson|0.44|0.21|0.32|0.34|
> |helpfulness|Somers' D|0.75|0.45|0.81|0.69|
> | |Pearson|0.64|0.41|0.64|0.72|
> |validity|Somers' D|0.83|0.73|0.85|0.82|
> | |Pearson|0.89|0.87|0.94|0.89|
> |
>
> Table 3. Comparison of Correlations Using Pearson and Somers' Methods Across Different Metrics and Models
>
> As seen from the results, there is a consistent pattern of correlation across different metrics and correlation methods, with no significant deviations. Regardless of the correlation method used, the relative rankings of the models' performance remain consistent, which further illustrates that our choice of Somers' D does not adversely impact the accuracy of our evaluation.
>
> We hope that this response addresses your comments and we appreciate your suggestions for improving the rigor of our work.
>
> &nbsp;
>
> ### **Response for additional baselines**
> As advised, we have already addressed the concerns about the baselines in our previous responses.
>
> **Factuality Baselines.**  As recommended, we have incorporated references [1,2] as factuality baselines and presented the results in Table 1.
>
> **Relevance Baselines.** Regarding Knowledge-F1[6] for relevance, we explained in our previous response that they primarily assess overlap between the response and its' grounding knowledge, which does not apply to the relevance evaluation for the generated knowledge.  As for the EM (Exact Match) and F1 Score used in [7], we've demonstrated these metrics respectively in Table 2 and Table 10 (in our paper).
>
> Thank you for your feedback.
>
> &nbsp;
>
> ### **Response for BERT-ranking model**
> As specified in Table 14 of our paper, we utilized the BERT Large model, which was trained on the TREC-CAR training set.
>
> This model was chosen based on its suitability for our task, as just discussed in the 'Motivation of using BERT-ranking-model' part. We aimed to capitalize on the model's existing capabilities rather than adapt it to our specific needs. This approach aligns with the "Why do we use off-the-shelf models" part, as mentioned in our prior response.
>
> Therefore, we decided to use the off-the-shelf BERT-ranking model as the component of our metric without any custom adaptations for measuring relevance. This strategy allowed us to maintain the broad applicability of our metrics.
>
> &nbsp;
>
> ### **Clarification for "the expected behaviour of informativeness"**
> We would like to clarify that our measure of informativeness is not strictly about the novelty of the text, but rather its ability to reduce the uncertainty introduced by a query.  We clarify this from two perspectives:
>
> **Conceptual Understanding of Informativeness.**
> In our study, we employ the conditional entropy of the acquired knowledge k given a query q. Conditional entropy can be understood as a measure of the uncertainty (or randomness) remaining in the 'knowledge' (k) once the 'query' (q) is known. In other words, it quantifies the 'uncertainty reduction' in the knowledge given a specific query.
>
> **Illustrative Example of Expected Behaviour.**
> To more intuitively understand the expected behaviour of 'informativeness', we consider a toy counterexample to illustrate a scenario where output may be novel but not informative:
>
> **Query:** What is the capital of the United States?
>
> **Acquired Knowledge:** Washington.
>
> In this case, the query is very straightforward, so the model's uncertainty is very low. The average probability of generating the "Washington" is high. Our informativeness score is 1 minus this average probability, thus resulting in a low informativeness score.
> Conversely, if the acquired knowledge was "China," the model's probability of generating this knowledge would certainly be low, leading to a high informativeness score.
>
> Similarly, for a very hard query where the model doesn't know the answer, any knowledge provided might have a high informativeness score because they all have low output probabilities from the model (since the model is uncertain about the answer).
>
> Therefore, the informativeness measure isn't just about novelty, but more about the ability of the text to reduce the uncertainty introduced by a query. We hope this clarifies the intended behaviour of our metric.
>
> &nbsp;
>
> ### **Response for typo**
> Apologies for the confusion.  The clearer representation of the equation should be as follows:
>
> $
> S_\texttt{help}(q, a, k, k_{1}^-, \cdots, k_{u}^-)
> =\max \left(0, 1 - \frac{\mathcal{L}(q,k,a)}{\frac{1}{u} \sum\nolimits_{i=1}^{u} \mathcal{L}(q,k_{i}^-,a)}\right)
> =\max \left(0, 1 - \frac{\log P(a|q, k)}{\frac{1}{u} \sum\nolimits_{i=1}^{u} \log P(a|q,k_{i}^-)}\right)
> $
>
> Thanks for pointing this out. We will also address other typos in our revised version.
>
> &nbsp;
>
>
> ### **Response for low sentence-coherence of DPR**
>
> Apologies if the coherence scores in Table 2 were not sufficiently explained. We explain it from two aspects: understanding the generally low coherence scores and explaining DPR's specific low score.
>
> **Understanding Generally Low Coherence Scores.** Our sentence coherence measure is based on the inverse of Perplexity (PPL), a common evaluation metric in natural language processing. Typically, PPL values for language models tend to range between 40-50, which results in lower sentence coherence scores when inverted. Despite the low absolute values, this does not impede our evaluation. The sentence coherence measure can effectively differentiate between models, providing a comparative assessment of their performance.
>
> However, it doesn't prevent us from using the sentence-level coherence measure to compare different large language models, as we only need to compare the relative values among different LLMs, not the absolute values.
>
> **Explaining DPR's Low Sentence-level Coherence Score.** The low coherence score for the Dense Passage Retrieval (DPR) and other models is mainly attributed to the nature of the knowledge passages they retrieve.
>
> Currently, the method [A3] for constructing the knowledge base involves truncating Wikipedia passages to a fixed length of 100 tokens. This often results in 'incomplete' passages, where the beginnings and ends of sentences are often cut off. This truncation can significantly impact the coherence of the text, leading to lower coherence scores.
>
> We hope this explanation provides a comprehensive understanding of the coherence scores in our study.
>
> &nbsp;
>
> ### **Response for qualitative examples.**
> We will adjust the DPR query to match the others and update Table 4 accordingly. And we appreciate your suggestion for a more clear qualitative analysis.
>
> &nbsp;
>
> ### **Response for pairwise correlation.**
> In Tables 5 and 6, we conducted pairwise correlations between two extrinsic and four intrinsic metrics across different datasets. To derive a more generalizable rule, we averaged the results from Language Model-based methods (LLMs) and compared them with the Dense Passage Retrieval (DPR), a retrieval-based method. This approach is designed to highlight the differences between the generative and retrieval methods.
>
> The main goal of our analysis was to ascertain which aspects of knowledge generation (intrinsic evaluation metrics) have the most substantial impact on the performance of downstream answer generation (extrinsic evaluation metrics). By correlating internal and external metrics, we can obtain a deeper understanding of the key factors that influence the performance of the external metrics. This method provides a quantifiable approach to pinpoint the most influential intrinsic properties. We hope this explanation offers a more comprehensive understanding of our correlation analysis and its objectives.
>
> &nbsp;
>
> ### **Clarification for '43.45'.**
> To clarify, our present values are relative improvements, not absolute ones. Specifically, we intended to convey that the improvement on the 'helpfulness' metric reached (0.209 - 0.146) / 0.146 = 43.15%, not 43.45% in the 'validity' metric. Our apologies for the confusion. Thanks for bringing this to our attention.
>
> &nbsp;
>
> ### **Response to Missing References.**
> Thank you for pointing that out. We will incorporate it in our revisions.
>
> &nbsp;
>
> ***
> We hope that our answers can address your concerns satisfactorily and improve the clarity of our contribution. We would be grateful if you could re-evaluate our paper. We look forward to receiving your further feedback.
>
> ***
> &nbsp;
>
> **References**
>
> [A1] Virginia Braun and Victoria Clarke. 2012. Thematic analysis. American Psychological Association.
>
> [A2] Alan Agresti. Analysis of ordinal categorical data, volume 656. John Wiley & Sons, 2010.
>
> [A3] Vladimir Karpukhin et. al 2020, Dense passage retrieval for open-domain question answering.

---

### Official Review · Reviewer_rAA7 · 2023-08-06

**Soundness:** 4

**Excitement:**

3: Ambivalent: It has merits (e.g., it reports state-of-the-art results, the idea is nice), but there are key weaknesses (e.g., it describes incremental work), and it can significantly benefit from another round of revision. However, I won't object to accepting it if my co-reviewers champion it.

**Paper Topic And Main Contributions:**

This paper tackles the problem of the evaluation of knowledge generated by LLMs. Prior evaluation methods only access the intrinsic quality of knowledge and overlook their extrinsic reliability on downstream tasks. Therefore, this paper proposes an evaluation framework named CONNER that considers six fine-grained aspects of intrinsic and extrinsic evaluations. The evaluation metrics consist of four intrinsic aspects (Factuality, Relevance, Coherence, and Informativeness) and two extrinsic aspects (helpfulness and Validity).

The main contribution is the evaluation framework that first includes extrinsic aspects and does not require human-labeled ground truth knowledge, and the insightful findings about knowledge generated by LLMs.

**Questions For The Authors:**

**Question A**: Whether the Helpfulness score is bias?

In Line 315, the intuitive baseline of Helpfulness is generating a response without any external knowledge instead of randomly sampled knowledge. The unrelated knowledge can negatively impact response generation. If so, the helpfulness metrics may be inflated. I am wondering whether these two types of baselines perform equally.

**Question B**: Could you describe the meaning of evidence, the reason to collect evidence, and the collection method In Line 347?

**Reasons To Accept:**

1. The reliability of knowledge generated by LLMs is important. This paper proposes a comprehensive evaluation framework to bridge the gap in knowledge evaluation and releases the code for further studies.

2. The paper conducts thorough evaluations and provides several interesting and instructive observations about generated knowledge of LLMs.

**Reasons To Reject:**

1. The evaluation framework is hard to generalize to languages besides English.

This framework relies on several assistant models, such as an NLI model, a ranking model, and a discourse coherence model. The availability of these assistant models poses a problem in other languages and hinders generalization.

2. Before applying the proposed metrics to three LLMs, this paper does not validate the effectiveness of the metrics.

Except for Section 6, it is essential to validate whether the metrics align with expectations when computed on the ground-truth knowledge, especially when the ground-truth knowledge is available in the used datasets.

**Reproducibility:**

3: Could reproduce the results with some difficulty. The settings of parameters are underspecified or subjectively determined; the training/evaluation data are not widely available.

**Reviewer Confidence:**

4: Quite sure. I tried to check the important points carefully. It's unlikely, though conceivable, that I missed something that should affect my ratings.

**Typos Grammar Style And Presentation Improvements:**

- According to Line 329, should the denominator of Equation (7) be the average rather than the sum?

---

> ### Author Rebuttal · Authors · 2023-08-28
>
> Thank you for your insightful and positive feedback! Your recognition of our work's importance, its role in bridging the knowledge evaluation gap, and its potential to facilitate future studies is deeply encouraging. We're also pleased that you acknowledged the thoroughness of our evaluations and the instructive observations they provide. Following this, we will sequentially address your comments and supplement our work with new experimental results.
>
>
> #### **Response for ​​"evaluation framework is hard to generalize to languages besides English".**
>
> Indeed, our framework relies on several common NLP components, possibly presenting generalization challenges for non-English languages, a dilemma faced by most NLP methodologies. However, model variants have been developed in other languages, such as Chinese [1][2][3], indicating potential for broader applications. Our work primarily aims to propose a general framework that can be applied to any language, so with the development of corresponding models in non-English languages, our method can be directly applied. We appreciate your insightful comments and will include further discussions regarding this issue in the Limitations section.
>
>
> #### **Response for "Before applying the proposed metrics to three LLMs, this paper does not validate the effectiveness of the metrics".**
>
> Actually, our paper indeed includes the 'validation of metric effectiveness'. We apologize for any confusion caused. For readability considerations, we initially placed the validation in **Chapter 6** and **Appendix.J.** The intention was to first present the exploratory findings and analyses, which we thought would hold more immediate value for readers. However, we acknowledge your insightful point about the importance of showcasing the metrics' effectiveness before applying them. We appreciate your suggestion and will make adjustments to enhance the rigour of our presentation in the revised version.
>
>
> #### **Response for "it is essential to validate whether the metrics align with expectations when computed on the ground-truth knowledge".**
>
> To better demonstrate the effectiveness of our metrics, we further supplemented our evaluation by computing our metrics on the ground-truth labels, after filtering out empty and noisy knowledge annotations in the data. The results are presented in Table 1.
>
> |Fact-con|non-verif|Fact-incon|Rel|sent-coh|para-coh|inform|helpful|valid|
> |:---:|:---:|:---:|:---:|:---:|:---:|:---:|:---:|:---:|
> |99.60%|0.40%|0.00%|0.957|0.0366|0.7803|0.9241|0.4326|81.32%|
> |
>
> Table 1: Evaluation metrics computed on ground-truth knowledge annotations
>
>
>
> The results were as expected — higher than all model results, except for coherence. The coherence discrepancy is due to two factors. First, the Perplexity (PPL) evaluation tends to favour text generated by language models [4]. Nonetheless, this does not prevent us from using it to gauge the fluency of knowledge generated by large language models. Secondly, the text generated by large language models indeed shows better paragraph coherence than human essays as they focus on transitions and are well-structured [5].
> We will include these results and corresponding explanations in the revised paper. We hope this response addresses your concerns.
>
>
>
> #### **Response for "whether these two types of baselines perform equally".**
>
> Generating a response without any external knowledge could be another viable baseline for Helpfulness. And we did consider the baseline of generating a response without any external knowledge but due to space constraints, we did not include this in the paper. We've supplemented additional experiments in NQ and detailed the motivation for choosing random knowledge as the baseline.
>
> **Supplementary experiments.**
> Notably, our experimental results in Table 2 show these two different implementations perform comparably. And, both implementations do not affect the relative ranking of model performance. We consider this consistency might be due to the minimal overlap between the given query and randomly sampled knowledge, leading the model to disregard the knowledge.
>
> |Metrics|DPR|Flan-T5|LLaMA|ChatGPT|Flan-T5|LLaMA|ChatGPT|
> |:---:|:---:|:---:|:---:|:---:|:---:|:---:|:---:|
> | |supervised||zero-shot|||few-shot||
> |randomknow|0.1236|0.0000|0.2191|0.1461|0.0000|0.2528|0.1966|
> |withoutknow|0.1086|0.0000|0.2057|0.1314|0.0000|0.2400|0.1827|
> |
>
> Table 2. Comparison of Helpfulness Scores on the NQ Dataset with Different Baselines
>
> **Motivation for why we use random knowledge.**
> The reason we chose to use random knowledge as a baseline is that our setup aligns more with the setting of open-book QA.  In such scenarios, some form of pre-acquired knowledge, regardless of its quality, is always provided as input.
>
> We truly appreciate your valuable comments about the Helpfulness score with a deep insight into our method. We will add the corresponding experiments and discussion in the final version, and we believe this additional analysis would strengthen our paper.
>
>
>
>
> #### **Response for "describe the meaning of evidence, the reason to collect evidence, and the collection method".**
>
> ● **Evidence.** In our scenario, "evidence" refers to Wikipedia Knowledge that is used to verify the correctness of generated knowledge. As an illustration, consider the generated knowledge 'The first Nobel Prize in Physics was awarded to Wilhelm Conrad Röntgen in 1901'. To confirm the correctness of this statement, which we aren't immediately sure of, we need to procure external knowledge. This might involve searching the Wikipedia page about 'the first Nobel Prize in Physics' to locate information that either validates or refutes the model's statement. In this context, corresponding knowledge on the Wikipedia page serves as our evidence.
>
> ● **Reason to collect evidence.** There are three primary reasons to collect evidence:
>
> (1)  We need to use reliable external knowledge as evidence to test the correctness of the generated knowledge, just like the example we have discussed above.
>
> (2)  The ground-truth knowledge in the dataset is not sufficient to fulfill this function (please refer to Table 10 for the NLI baseline result). Ideally, the real 'ground truth' should be annotated in response to model-generated knowledge. However, the 'ground-truth knowledge' in the dataset typically corresponds to annotations made against pre-existing human answers, which may not align with the content of model-generated knowledge. For instance, it might only contain the name of the Nobel Prize laureate (information from the human answer), without encompassing the year of the award (additional information from the model answer). This discrepancy underscores the necessity for a more comprehensive evidence collection to ensure the thorough and accurate evaluation of the generated knowledge.
>
> (3)  In real-world scenarios, it's not viable to have ground-truth knowledge readily available, making the collection of evidence a necessary step.
>
> ● **Collection Method.** To ensure comprehensive coverage of the knowledge generated by the model, the following approach is employed. Firstly, the generated knowledge is decomposed into individual sentences. Each sentence is then utilized as a query, with Wikipedia serving as the knowledge base, and ColBERTv2 acting as the retrieval model. The objective is to retrieve several of the most relevant evidence related to each sentence in order to verify its correctness. Once all the sentences have been validated, the results are aggregated using formula 1 to determine the overall correctness of the entire knowledge.
>
> We thank the reviewer for such valuable comments, we will detail them in the revised version.
>
>
> #### **Response for typo.**
> Thank you for pointing this out. We will correct them in our revised version.
>
> &nbsp;
> ***
>
> We hope that our answers can address your concerns satisfactorily and improve the clarity of our contribution. We would be grateful if you could re-evaluate our paper. We look forward to receiving your further feedback.
>
> ***
> &nbsp;
>
> **References**
>
> [1] Hai Hu et al. 2020, OCNLI: Original Chinese Natural Language Inference
>
> [2] Xiaohui Xie et al. 2023, A large-scale Chinese Benchmark for Passage Ranking
>
> [3] Guimin Huang et al. 2017, A discourse coherence model for analyzing Chinese students' essay
>
> [4] Yafu Li et al. 2023, Deepfake Text Detection in the Wild
>
> [5] Biyang Guo et al. 2023, How Close is ChatGPT to Human Experts? Comparison Corpus, Evaluation, and Detection

---

### Meta-Review · Area_Chair_4muX · 2023-09-12

**Recommendation:** 5

**Metareview:**

This paper addresses the problem of evaluating the knowledge extracted from LLMs. The proposed evaluation metrics consist of four intrinsic aspects (Factuality, Relevance, Coherence, and Informativeness) and two extrinsic aspects (Helpfulness and Validity). The authors propose an automatic metric for each aspect and evaluate generated knowledge from 3 LLMs (Flan-T5, Llama and ChatGPT) and text from retrieval system (DPR) across using these metrics. Their results indicate that LLM generate text are less factual but more relevant than retrieved text, and that the higher relevance is ultimately important for downstream tasks.
The reviewers appreciate the importance of verifying the information generated by LLMs and the interesting analysis provided by the paper.

The reviewers initially had some concerns specifically with respect to the validity / efficacy of the metrics, especially the informativity metric. However, these concerns seem to have been largely resolved in the discussion with the authors such that it seems likely that the authors can address the concerns in the revision of the paper.

---

### Decision · Program_Chairs · 2023-10-07

**Decision:**

Accept-Main

**Comment:**

This paper addresses the problem of evaluating the knowledge extracted from LLMs. The proposed evaluation metrics consist of four intrinsic aspects (Factuality, Relevance, Coherence, and Informativeness) and two extrinsic aspects (Helpfulness and Validity). The authors propose an automatic metric for each aspect and evaluate generated knowledge from 3 LLMs (Flan-T5, Llama and ChatGPT) and text from retrieval system (DPR) across using these metrics. Their results indicate that LLM generate text are less factual but more relevant than retrieved text, and that the higher relevance is ultimately important for downstream tasks.
The reviewers appreciate the importance of verifying the information generated by LLMs and the interesting analysis provided by the paper.

The reviewers initially had some concerns specifically with respect to the validity / efficacy of the metrics, especially the informativity metric. However, these concerns seem to have been largely resolved in the discussion with the authors such that it seems likely that the authors can address the concerns in the revision of the paper.